# Seismic evidence of the COVID-19 lockdown measures: a case of study from Eastern Sicily (Italy)

Andrea Cannata[1,2], Flavio Cannavò[2], Giuseppe Di Grazia[2], Marco Aliotta[2], Carmelo Cassisi[2], Raphael S. M. De Plaen[3], Stefano Gresta[1], Thomas Lecocq[4], Placido Montalto[2], Mariangela Sciotto[2]

[1]Dipartimento di Scienze Biologiche, Geologiche e Ambientali, Università Degli Studi di Catania, Catania, Italy.
[2]Istituto Nazionale di Geofisica e Vulcanologia, Osservatorio Etneo, Catania, Italy.
[3]Centro de Geociencias, Universidad Nacional Autónoma de México, Campus Juriquilla, Querétaro, Mexico.
[4]Seismology-Gravimetry, Royal Observatory of Belgium, Avenue circulaire 3, 1180 Brussels, Belgium.

*Correspondence to*: Andrea Cannata (andrea.cannata@unict.it)

**Abstract.** During the COVID-19 pandemic, most countries put in place social interventions, restricting the mobility of citizens, to slow the spread of the epidemic. Italy, the first European country severely impacted by the COVID-19 outbreak, applied a sequence of progressive restrictions to reduce human mobility from the end of February to mid-March 2020. Here, we analysed the seismic signatures of these lockdown measures in the densely populated Eastern Sicily, characterised by the presence of a permanent seismic network used for earthquake and volcanic monitoring. We emphasize how the anthropogenic seismic noise decrease is visible even at stations located in remote areas (Etna and Aeolian Islands) and the amount of this reduction (reaching ~50-60%), its temporal pattern and spectral content are strongly station-dependent. Concerning the latter, we showed that on average the frequencies above 10 Hz are the most influenced by the anthropogenic seismic noise. We found similarities between the temporal patterns of anthropogenic seismic noise and human mobility, as quantified by the mobile phone-derived data shared by Google, Facebook and Apple, as well as by ship traffic data. These results further confirm how seismic data, routinely acquired worldwide for seismic and volcanic surveillance, can be used to monitor human mobility too.

## 1. Introduction

During the end of 2019, several cases of pneumonia, due to the novel coronavirus SARS-CoV-2, were identified in the city of Wuhan, China (Wang et al., 2020a). The disease due to this coronavirus, called COVID-19, then rapidly spread from China to other areas, as a pandemic wave (as declared by the World Health Organisation, WHO, in March 2020) currently affecting 216 countries with almost 14,300,000 confirmed cases (at the time of writing 21 July 2020; WHO, 2020). COVID-19 is considered the most severe global health crisis of our time and the greatest challenge human beings have faced since world war II (WHO, 2020).

While huge efforts are being made to find pharmacological cures to heal the sick and to stop the spread of the disease (e.g. Cannata et al., 2020a; Graham, 2020; Wang et al., 2020b), most countries worldwide put in place social interventions, consisting of restricting the mobility of citizens, aimed at slowing and mitigating the epidemic (Pepe et al., 2020). Italy was the first country in Europe to be severely impacted by the COVID-19 pandemic wave at the end of February – onset of March 2020. Hence, Italy was also the first European country to apply a sequence of progressive restrictions to reduce both human

mobility and human-to-human contacts. Restrictions were first implemented on 23 February 2020 in some regions of Northern Italy (Lombardy, Emilia-Romagna, Veneto, Friuli-Venezia Giulia, Piedmont, and Autonomous Province of Trento). On 11 March, the entire country was put under lockdown (Gatto et al., 2020) up to May, when the restrictions were gradually lifted.

  To provide information about the effectiveness of the quarantine measures during COVID-19 emergency, Apple, Facebook and Google made available mobility data, mostly based on mobile phone locations, for almost every country in the world

(Apple, 2020; Facebook, 2020; Google, 2020). At the same time, different studies showed the effectiveness of seismic noise monitoring as a tool to quantify human activity and its changes over time (Dias et al., 2020; Hong et al., 2020; Lindsey et al., 2020; Lecocq et al., 2020; Poli et al., 2020). Indeed, the Earth is continuously vibrating due to a wide spectrum of elastic energy sources including tectonic forces (Stein and Wysession, 2003), volcanic processes (Chouet and Matoza, 2013), ocean (Cannata et al., 2020b) and human (Diaz et al., 2017) activity. As for the latter, it typically generates a high-frequency

continuous signal (> 1 Hz), called anthropogenic or cultural seismic noise, associated with phenomena such as traffic, construction, industrial operations and mining (Diaz et al, 2017; Hong et al., 2020). Recent papers identified clear seismic signatures of the lockdown measures applied by different countries (Lindsey et al., 2020; Lecocq et al., 2020; Piccinini et al., 2020; Poli et al., 2020; Xiao et al., 2020). For instance, Lindsey et al. (2020) showed a 50% decrease in vehicle count in Palo Alto area (California) immediately following the lockdown order by using fiber-optic distributed acoustic sensing connected

to a telecommunication cable. Lecocq et al. (2020) performed a global-scale analysis of anthropogenic seismic noise using hundreds of seismometers located around the world, which evidenced how the 2020 lockdown period has produced the longest and most dramatic global anthropogenic seismic noise reduction on record. Poli et al. (2020) and Piccinini et al. (2020) analysed the anthropogenic seismic noise reduction following the lockdown for the Northern Italy area and its socio-economic implications.

In this work, we analyse the seismic signatures of the lockdown measures in the highly populated Eastern Sicily (Italy), which benefits from the presence of a dense permanent seismic network used for both seismic and volcanic monitoring. Hence, this is the first study to show the effects of uniform lockdown measures in the seismic noise acquired by a so dense seismic network. In particular, we investigate the decrease in the anthropogenic seismic noise amplitude, characterise its spectral content, and compare the observed changes with mobility data.


## 2. Materials and methods

### 2.1 Data

The seismic data was recorded by 18 stations, located on the Eastern part of Sicily and belonging to the seismic permanent network, run by Istituto Nazionale di Geofisica e Vulcanologia, Osservatorio Etneo (INGV-OE) (Table 1, Figures 1 and A1). These stations, selected both for the good data continuity and the even spatial distribution on the investigated area, are equipped with broadband (40 s cutoff period), 3-component Trillium Nanometrics™ seismometers, acquiring at a sampling rate of 100 Hz. The stations are installed in shallow vaults (depth ~1.5 m) made of concrete, and show very different site conditions in

terms of possible sources of anthropogenic seismic noise; some are close to towns, highways or industrial plants, others near agricultural areas, others in small islands or on the flanks of Mt. Etna volcano (see Table 1). The analysed time interval was 1 November 2019 – 23 May 2020.

**Table 1.** List of the seismic stations used in this work, with information about location and site conditions in terms of possible sources of anthropogenic seismic noise.

| Station name | Latitude (degree) | Longitude (degree) | Altitude (m a.s.l.) | Site conditions |
|:---:|:---:|:---:|:---:|---|
| AIO | 37.9713 | 15.233 | 794 | close to small towns and a few roads |
| CAGR | 37.622 | 14.4999 | 548 | close to small towns, a few roads and agricultural areas |
| EFIU | 37.7896 | 15.2103 | 97 | close to towns, roads and highway |
| EMFS | 37.7196 | 14.9979 | 2507 | on the flank of Mt. Etna, close to country roads |
| ESAL | 37.7551 | 15.1345 | 768 | close to small towns and a few roads |
| ESML | 37.6181 | 14.8794 | 408 | close to towns and roads |
| ESPC | 37.6925 | 15.0274 | 1655 | on the flank of Mt. Etna, close to country roads |
| HAGA | 37.2858 | 15.1552 | 176 | close to town, roads and industrial plants |
| HLNI | 37.3486 | 14.8719 | 133 | close to towns, roads, agricultural areas and industrial plants |
| HPAC | 36.7085 | 15.0372 | 70 | close to small towns, roads and agricultural areas |
| HSRS | 37.0928 | 15.222 | 100 | close to towns, roads, highway and industrial plants |
| IFIL | 38.5642 | 14.5753 | 269 | on an island, close to a port |
| ILLI | 38.4457 | 14.9482 | 277 | on an island, close to a port |
| ISTR | 38.7867 | 15.1918 | 114 | on an island, close to a port |
| IVCR | 38.4096 | 14.961 | 172 | on an island, close to a port |

| | | | | |
|---|---|---|---|---|
| MSFR | 38.0339 | 14.5916 | 723 | close to small towns and a few roads |
| MSRU | 38.2639 | 15.5083 | 401 | close to small towns and a few roads |
| MUCR | 38.043 | 14.8739 | 1042 | close to small towns and a few roads |

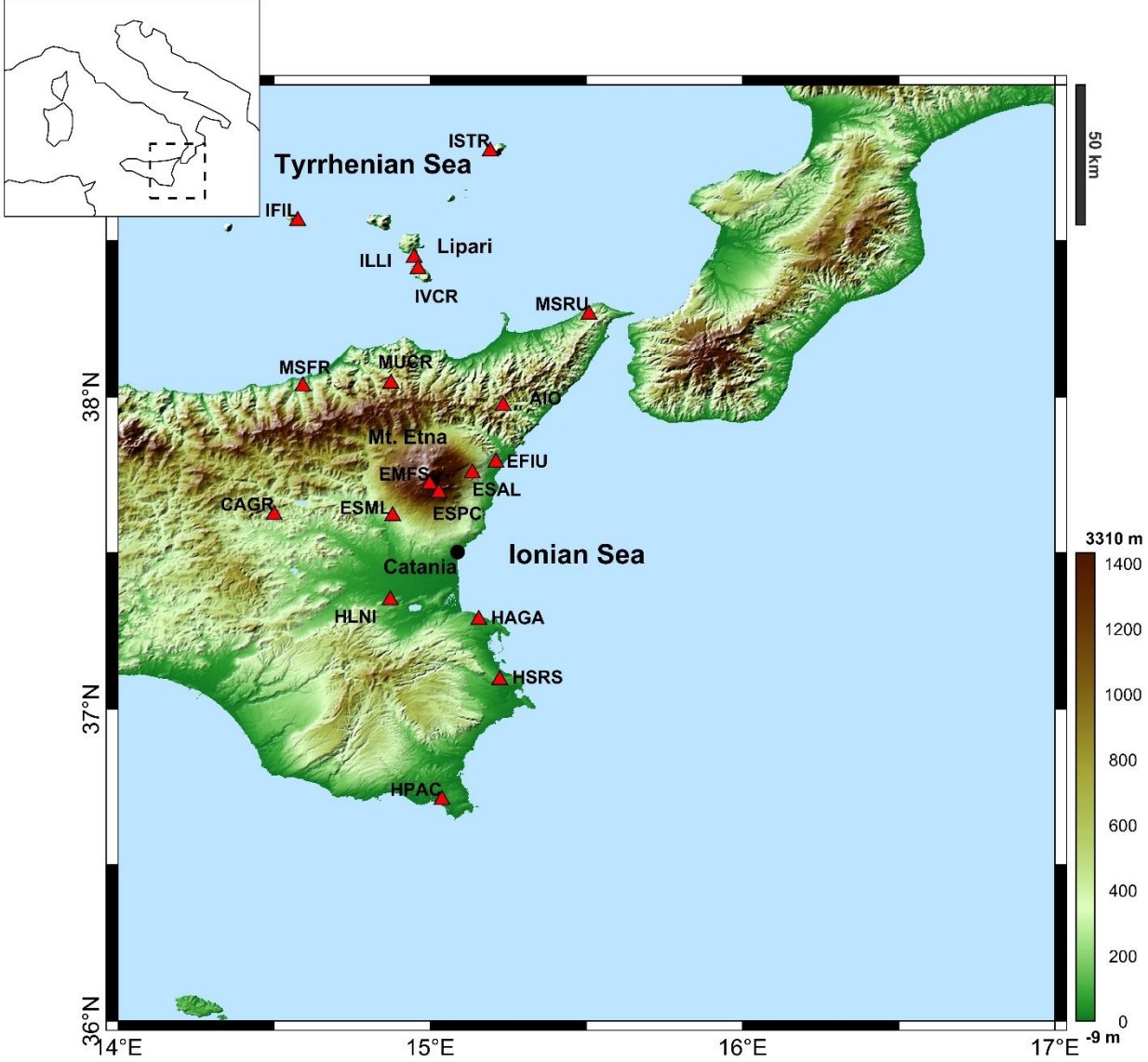


**Figure 1. Map of the Eastern Sicily with the location of the seismic stations (red triangles) used in this work. The top left panel shows where the area is located in Italy. Map was drawn by using NASA Shuttle Radar Topography Mission (SRTM) data and readhgt.m Matlab® function (Beauducel, 2020).**

## 2.2 Spectral and amplitude analysis

Both spectral and amplitude analyses were carried out to characterise the temporal variations of the seismic noise features. As for the former, the daily spectra of the vertical component of the seismic signal were calculated by using Welch's method (Welch, 1967) with windows of 81.92 seconds. All the daily spectra were gathered and visualized as spectrograms, with time on the x-axis, frequency on the y-axis, and power spectral density (PSD) indicated by a color scale (Figure 2). It is worth noting that the sharp decrease in the spectral amplitude, evident at frequencies above 40 Hz, is due to the digitizer anti-aliasing filter. Hence, the following analyses have been performed at frequencies up to 40 Hz.

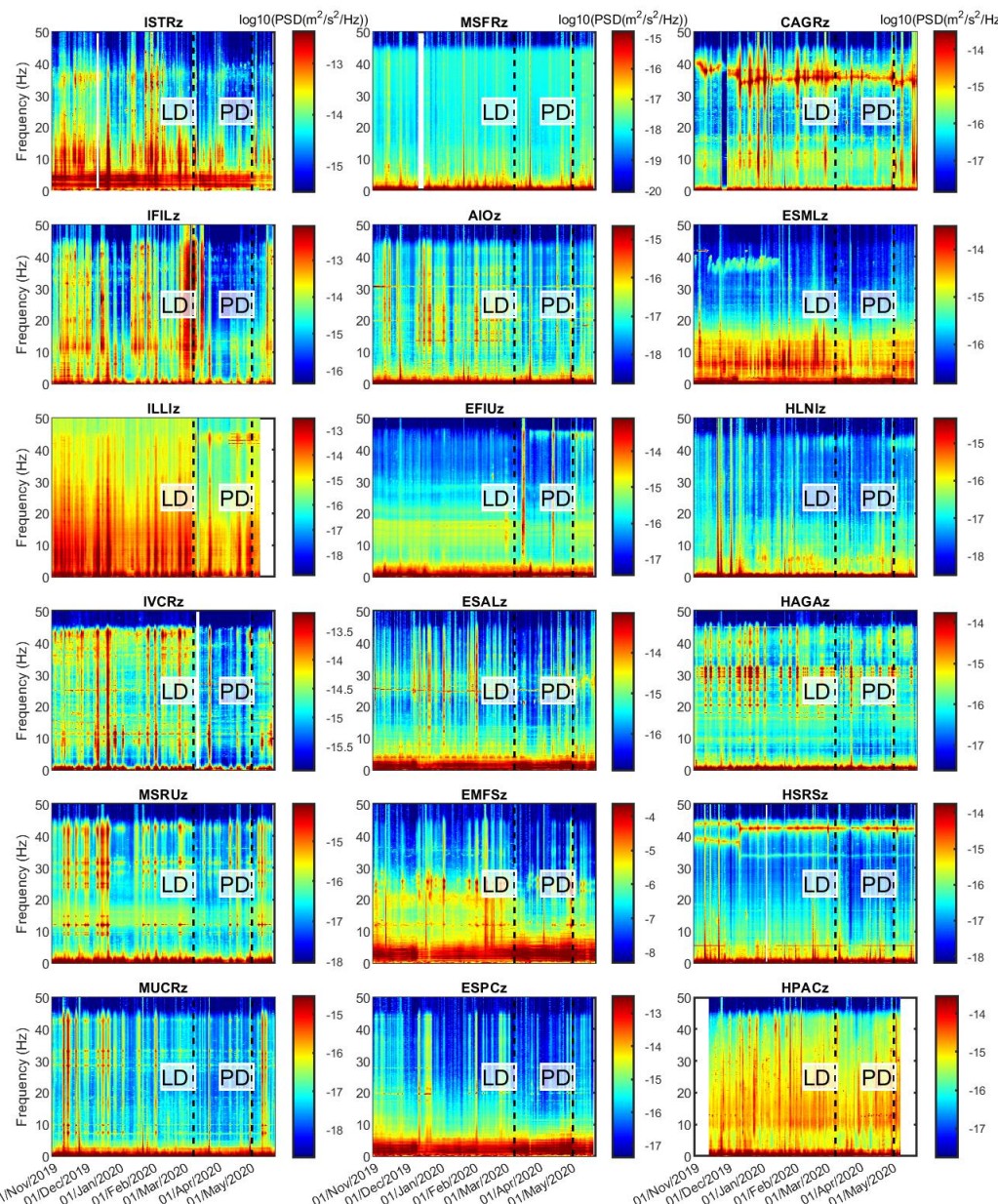

**Figure 2. Spectrograms of the vertical component of the seismic signals. The vertical dashed lines and the labels "LD" and "PD" indicate the times when the national lockdown measures were implemented in Italy (11 March 2020) and when first Presidential Decree, slightly releasing the lockdown measures, was issued (4 May 2020), respectively. The stations are sorted by decreasing latitude from upper left to bottom right.**

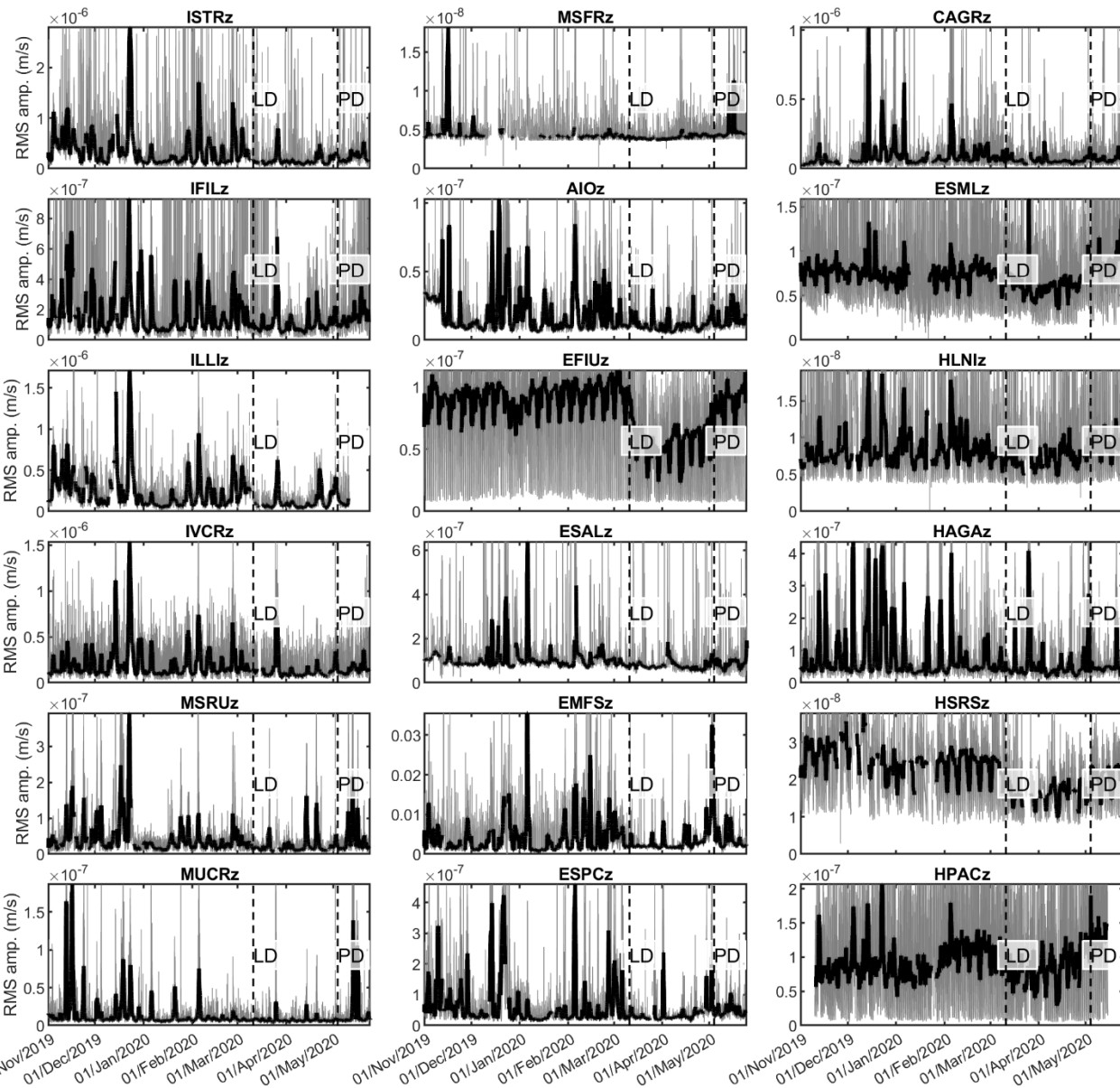

**Figure 3. RMS amplitude time series of the vertical component of the seismic signals, filtered in the band 10-40 Hz (grey line) and corresponding moving median computed on 1-day-long sliding windows (thick black line). The vertical dashed lines and the labels "LD" and "PD" indicate the times when the national lockdown measures were implemented in Italy (11 March 2020) and when first Presidential Decree, slightly releasing the lockdown measures, was issued (4 May 2020), respectively. The stations are sorted by decreasing latitude from upper left to bottom right.**

Concerning the amplitude analysis, the time series of the root mean square (RMS) amplitude of the seismic signal, filtered in
the band 10-40 Hz, were obtained on 15-minute-long sliding windows (Figure 3). This frequency band was chosen because,
as it will be shown in the section 3, it is the most influenced band by the anthropogenic seismic noise. To visually show the
general temporal pattern of the seismic noise amplitude in Eastern Sicily, the RMS amplitude time series were averaged on 3-
day-long sliding windows, normalised, gathered and represented by a colored checkerboard plot (Figure 4). To make the
changes of the noise background level as clear as possible in the checkerboard plot, the normalization was performed by: i)
setting all the values greater than the 90th percentile, equal to the 90th percentile, ii) subtracting the minimum value, and iii)
dividing by the maximum value. In addition, the percentage change in seismic RMS amplitude in the band 10-40 Hz in the
period 11 March - 11 April 2020 was calculated by using the RMS amplitude during 20 January - 20 February 2020 as baseline
(Figure 5).

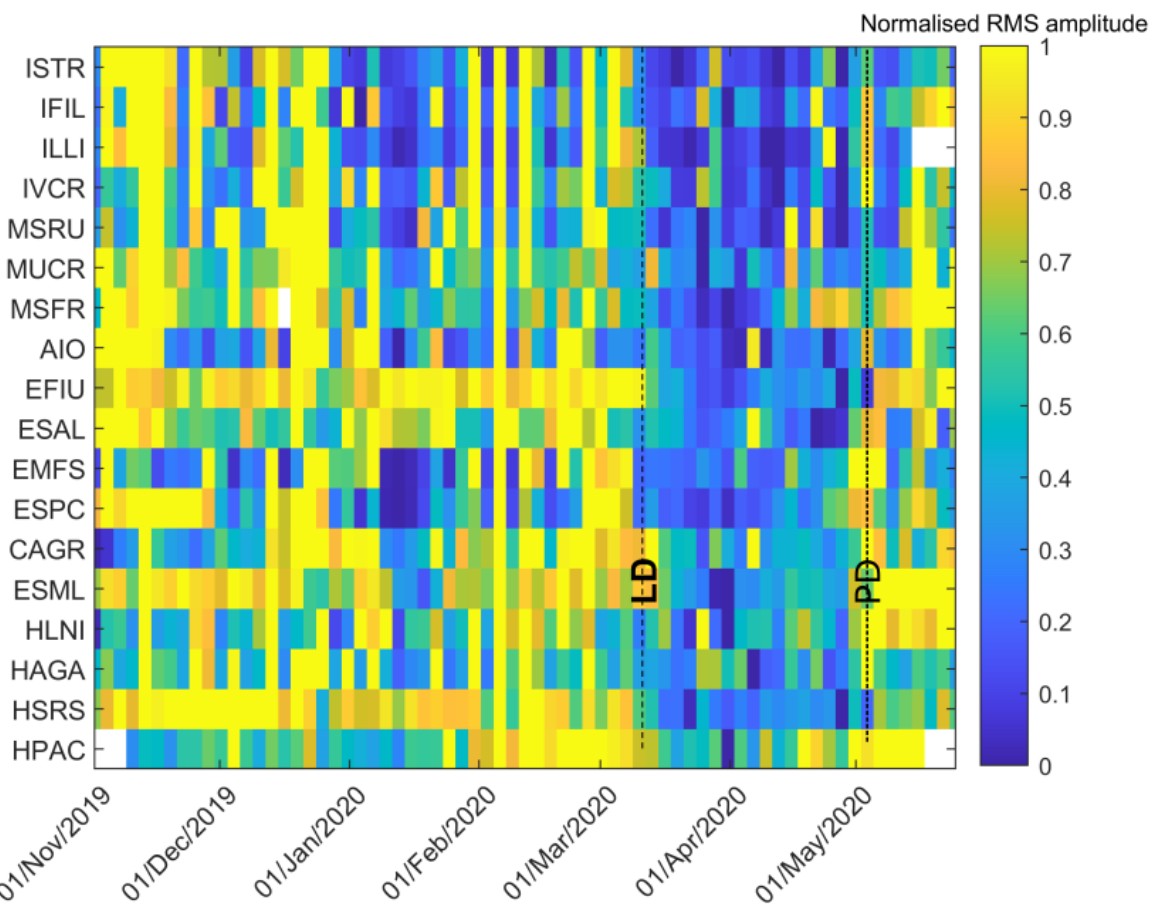

**Figure 4. Temporal changes of the normalised seismic RMS amplitude at 18 seismic stations sorted by decreasing latitude (see labels
close to the left y-axis). Data gaps are coloured white. The vertical dashed lines and the labels "LD" and "PD" indicate the times
when the national lockdown measures were implemented in Italy (11 March 2020) and when first Presidential Decree, slightly
releasing the lockdown measures, was issued (4 May 2020), respectively.**

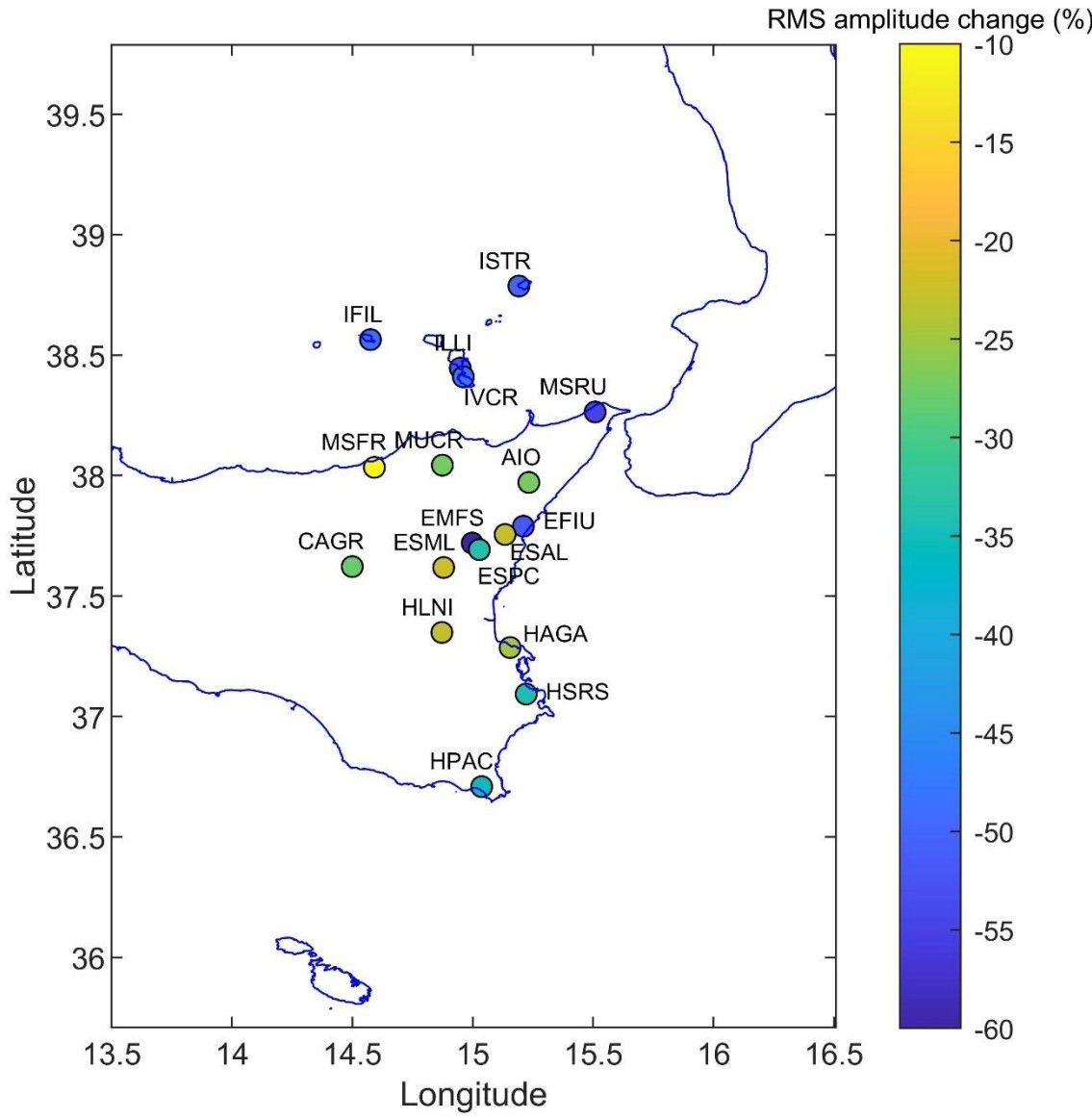


**Figure 5. Percent change of seismic RMS amplitude in the band 10-40 Hz during the period 11 March - 11 April 2020 (right after the lockdown measures entered in force) with respect to the interval 20 January - 20 February 2020.**

Finally, to highlight the frequency band showing the most evident amplitude changes due to the lockdown, two 20-day-long

time windows, extracted before (1-20 February) and during (11 March – 1 April) the lockdown, were considered. Two average

spectra, representing the seismic spectral content before and during the lockdown, were computed on the two windows, and,

successively, a ratio between them was calculated (Figures 6 and 7). All the ratios were stacked and the result is shown in Figure 8.

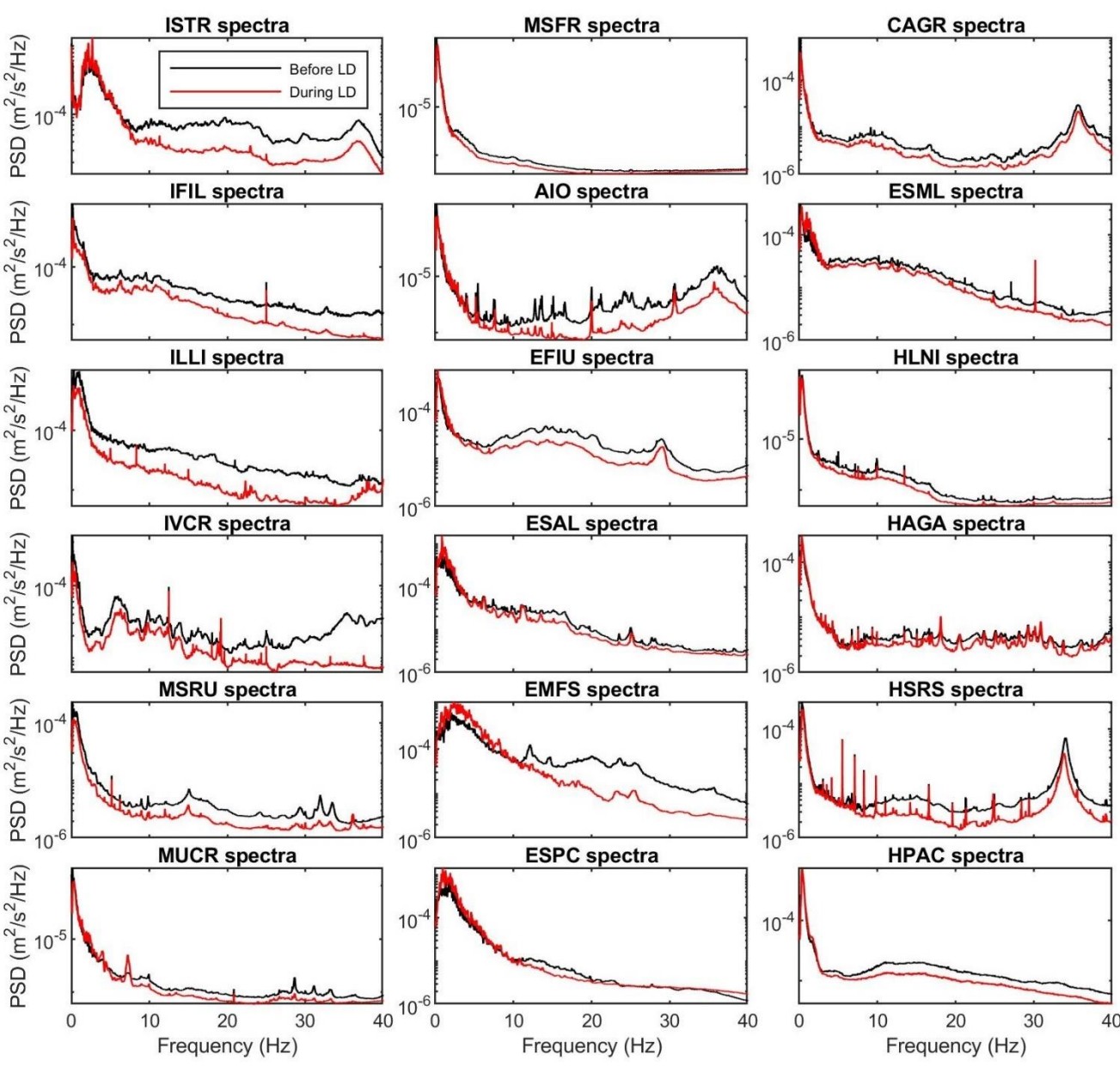


**Figure 6. (a) Spectra of the vertical component of the seismic signals recorded during two 20-day-long time windows, extracted before (black line; 1-20 February) and during (red line; 11 March – 1 April) the lockdown. The stations are sorted by decreasing latitude from upper left to bottom right.**

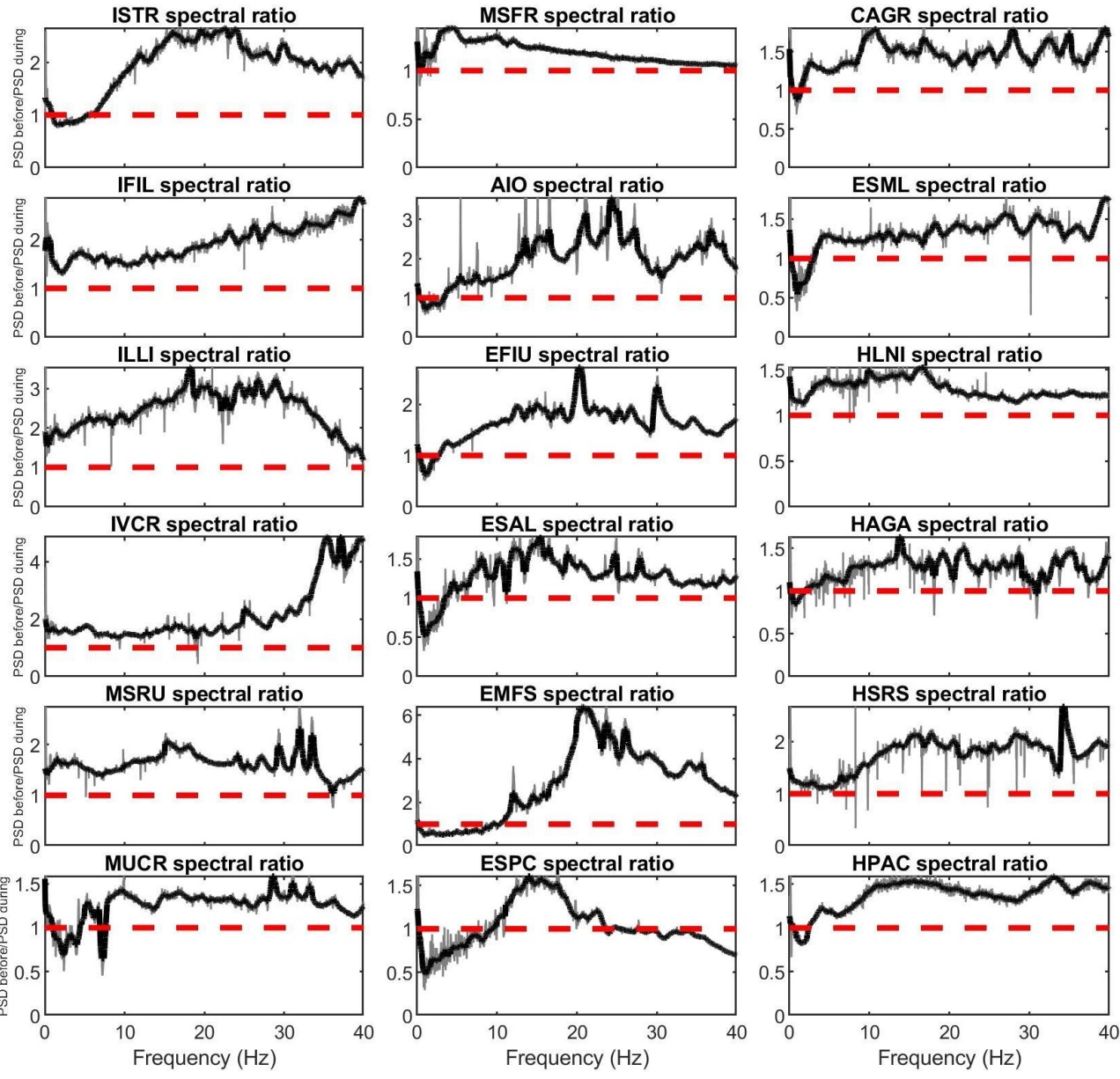

**Figure 7. Ratios between the spectra of the vertical component of the seismic signals recorded during two 20-day-long time windows, extracted before (1-20 February) and during (11 March – 1 April) the lockdown (grey lines; see Figure 6) and corresponding moving median over 0.6 Hz (black thick line). The red horizontal dashed lines indicate ratio values equal to 1. The stations are sorted by decreasing latitude from upper left to bottom right.**

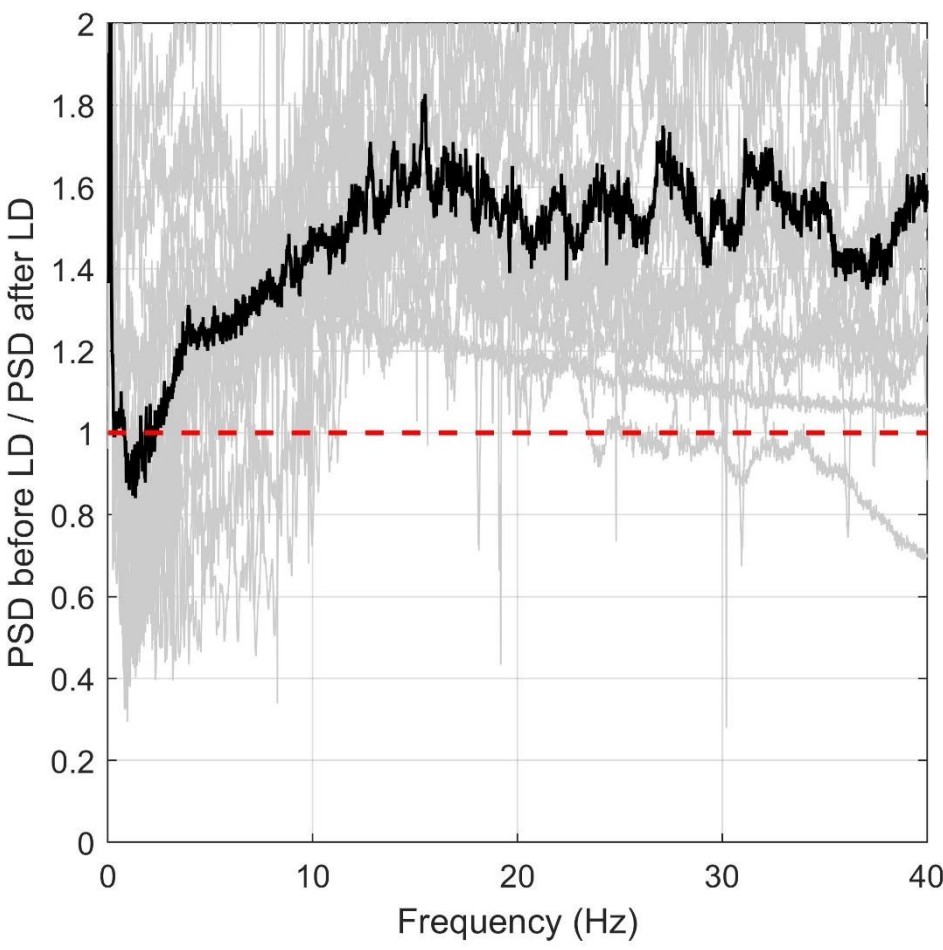

**Figure 8. Ratios between the spectra computed before and during the lockdown (grey lines; see Figure 7) and the corresponding stacked ratio (black line). The red horizontal dashed line indicates a ratio value equal to 1.**

## 2.3 Comparison with mobility data

Google, Apple and Facebook made human mobility data available for almost every country worldwide as support for public
health policy during the COVID-19 crisis. For our study area, Google and Facebook provided aggregated data for the Sicily
region, Apple shared data for Catania City, which is the main city in Eastern Sicily (Figure 1). The Google community mobility
corresponds to the percentage of change relative to a baseline defined as the median value of the corresponding day of the
week, during the period 3 January – 6 February 2020. The data are structured in categories to group some of the places with
similar characteristics: grocery stores and pharmacy, parks, transit stations, retail and recreation places, residences and
workplaces (Google, 2020). Apple shared information about the percent change of public's walking and driving compared with

the baseline value from the 13 January (Apple, 2020). Finally, Facebook provided data regarding the human movement percent changes measured throughout March, April, and May 2020 relative to a baseline value in February (Facebook, 2020).

To understand how much the seismic noise could reflect the society mobility level, a preliminary visual comparison between time series of the seismic RMS amplitude and the corresponding above mentioned community mobility data was performed (Figure 9a). Then, to quantify the similarity, a correlation analysis was performed. In place of using the more common Pearson correlation coefficient, we made use of the Spearman correlation coefficient, allowing to compare series which do not have a normal distribution, as well as to explore nonlinear relationships (e.g. Craig et al., 2016; Cannata et al., 2019). To identify the frequencies that better correlate with the human mobility data, we performed the correlation analysis between all the time series of seismic RMS amplitudes, filtered in narrow bands (bandwidth = 1Hz, 4th order Butterworth filter; Lyons, 2004) around the integer frequencies between 1 and 40 Hz, and the community mobility datasets provided by Google, Apple and Facebook. Successively, we computed an average value of the Spearman correlation coefficients of all the stations per each frequency band (Figure 9b). In addition, to show how the correlation changes at the different considered stations, the Spearman correlation coefficient was calculated between the seismic RMS amplitude time series of each station in the 10-40 Hz range and the community mobility datasets (Figure 10). To verify if the obtained Spearman correlation coefficients are significantly different from zero or not (null hypothesis), the t-test was performed and the p-value (probability value) was calculated (Figure A3). p-values lower than the significance level of 0.05 were considered sufficient to reject the null hypothesis. Such a value, which means that the probability that the result of the statistical test is due to chance alone is less than 5%, is a commonly accepted threshold for this statistic test (e.g. Anthony et al., 2017).

In addition, since some seismic stations are located on the Aeolian Islands (ISTR, IFIL, ILLI, IVCR), we took into account ship traffic data from Lipari port (~2.7 km from ILLI station; see Figure 1), provided by FleetMon.com, containing information about the daily number of port calls of all the boats, as well as each boat category (passenger, highspeed, tanker, cargo and yacht) and their gross tonnage. All these data were compared with the seismic RMS amplitude time series of ILLI station in the 10-40 Hz band again by Spearman correlation coefficient (Figure 11a-d). As the only parameters showing p-values lower than 0.05 were the overall daily gross tonnage and the daily number of port calls of tanker, the frequency dependence analysis shown above was repeated on these two time series (Figure 11e).

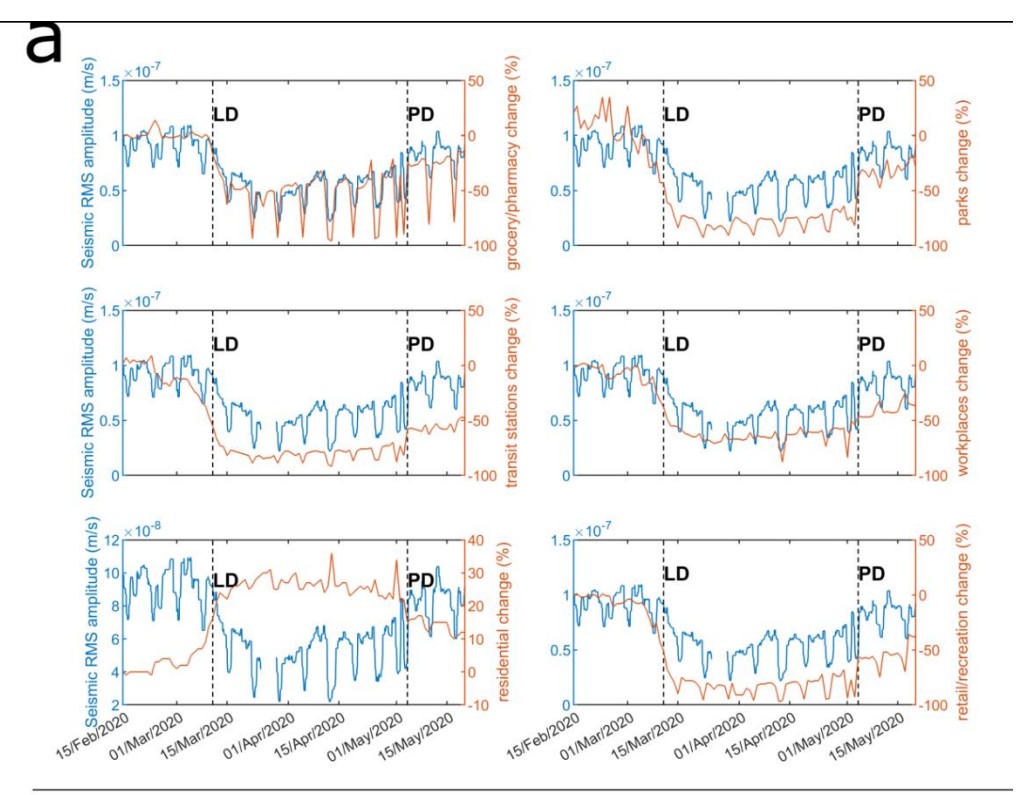

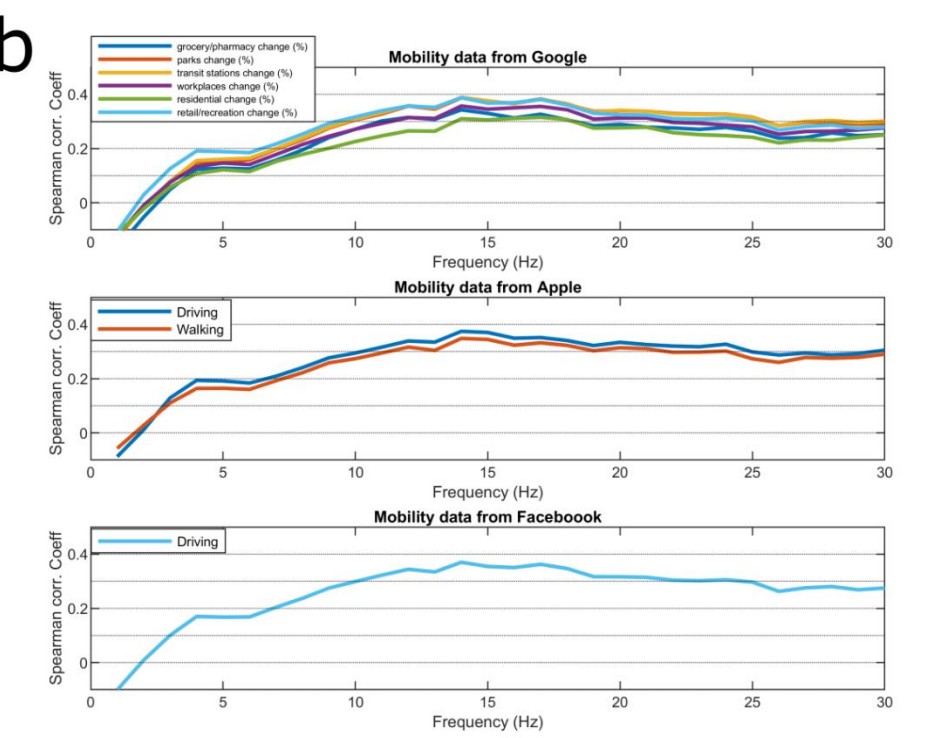

**Figure 9. (a) Time series of RMS amplitude of the vertical component of the seismic signal recorded by EFIU station and filtered in the band 10-40 Hz (blue line) and the different categories of human mobility as provided by Google (red line). The vertical dashed lines and the labels "LD" and "PD" indicate the times when the national lockdown measures were implemented in Italy (11 March 2020) and when first Presidential Decree, slightly releasing the lockdown measures, was issued (4 May 2020), respectively. (b) Spearman correlation coefficient, calculated between seismic RMS amplitude at the different stations and the mobility parameters, as a function of the frequency band of the seismic noise. The Spearman correlation coefficient obtained for the residential change (top plot in (b)) was multiplied by -1 to make it positive.**

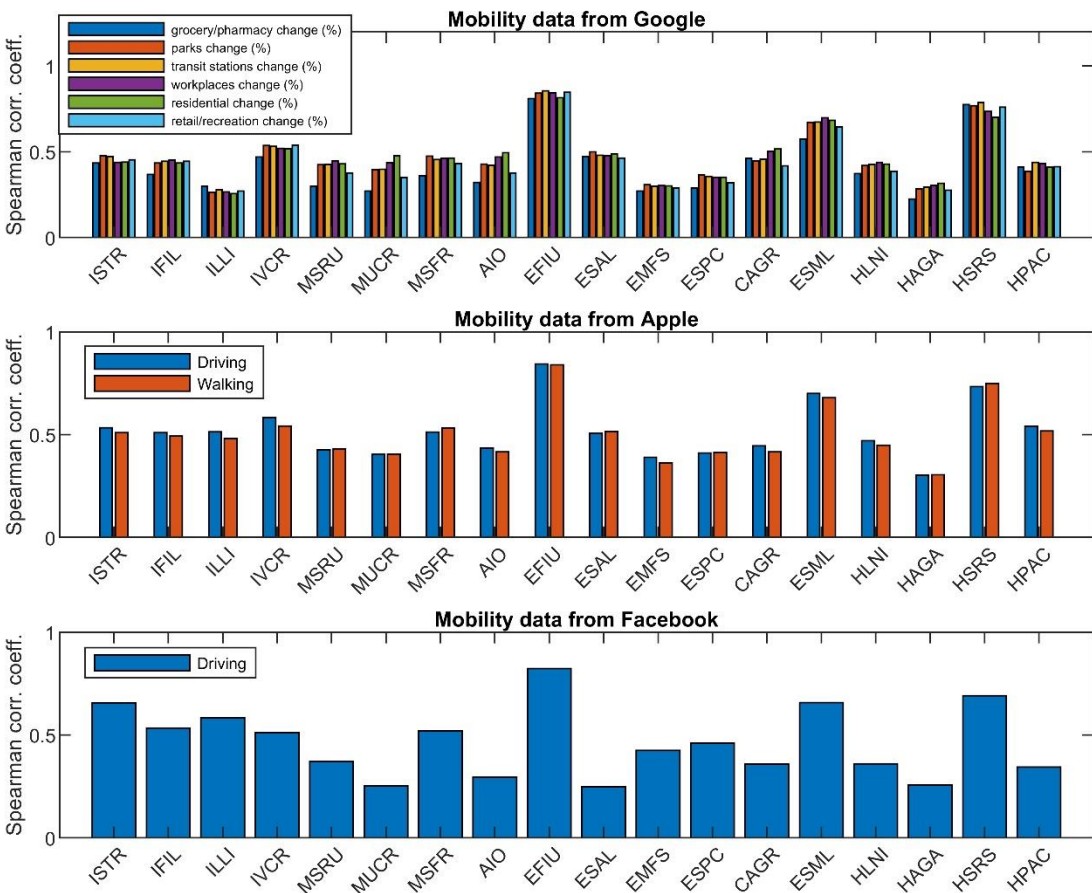

**Figure 10. Spearman correlation coefficient calculated between seismic RMS amplitude at the different stations in the band 10-40 Hz and the mobility parameters, as provided by Google, Apple and Facebook. The Spearman correlation coefficient obtained for the residential change (see top plot) was multiplied by -1 to make it positive.**

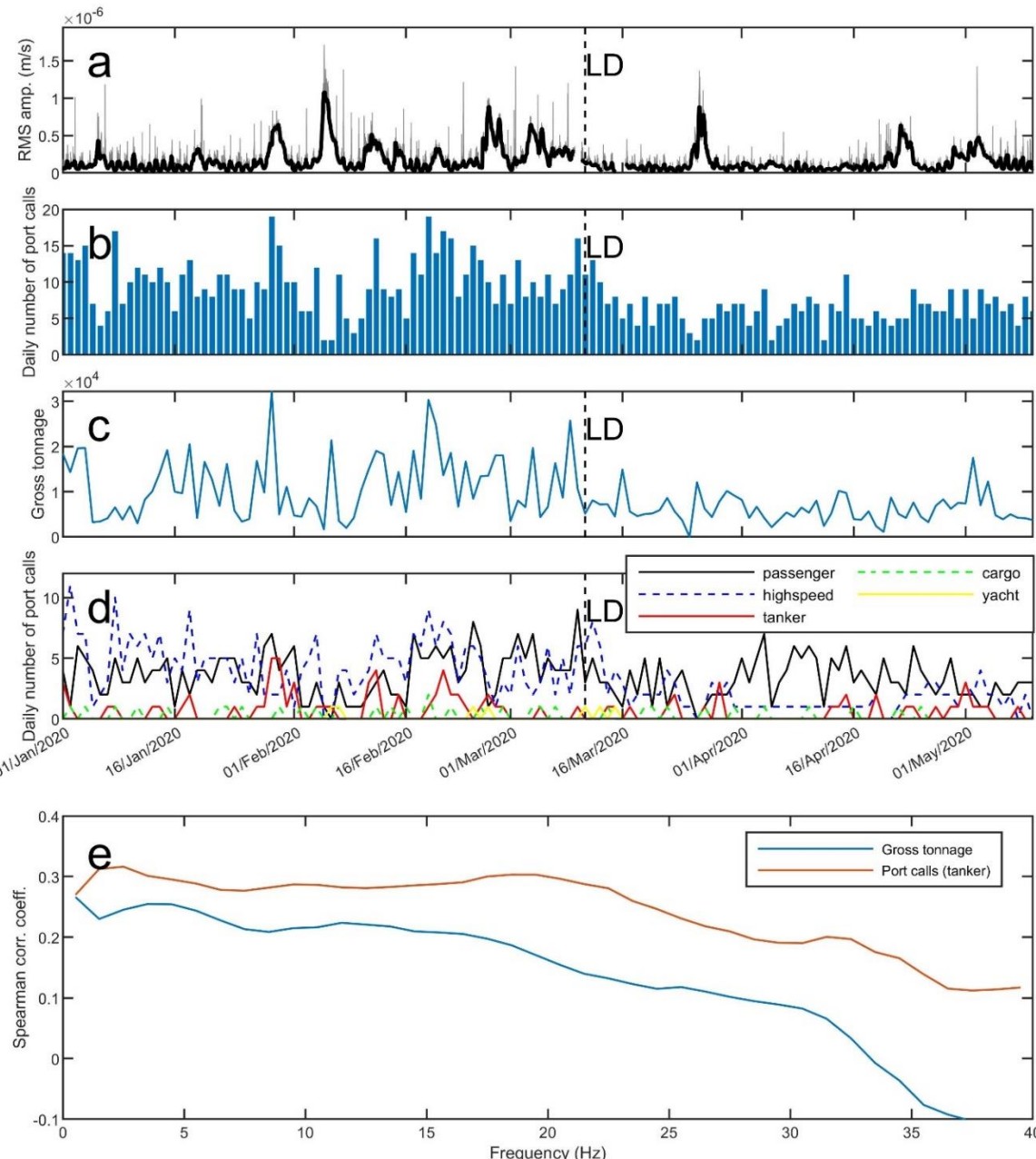

**Figure 11. (a)** RMS amplitude time series of the vertical component of the seismic signal, recorded by ILLI station and filtered in the band 10-40 Hz (grey line) and corresponding moving median computed on 1-day-long sliding windows (thick black line). **(b)** Daily number of port calls in the Lipari port. **(c)** Daily overall gross tonnage in the Lipari port. **(d)** Daily number of port calls in the Lipari port, separated on the basis of the boat category. **(e)** Spearman correlation coefficient, calculated between seismic RMS amplitude at ILLI and two ship traffic parameters (see the legend in the upper right corner), as a function of the frequency band of the seismic noise. The vertical dashed line and the label "LD" in (a-d) indicate the time when the national lockdown measures were implemented in Italy (11 March 2020).

## 3. Results and discussion

The seismic data, collected by 18 stations located in Eastern Sicily during 1 November 2019 – 23 May 2020, was analysed.

The spectrograms show a wide variety of spectral features, as well as their variability over time (Figure 2). Some stations show broad spectra with significant amplitude up to 30-40 Hz (such as ISTR and IVCR), others narrower spectra with almost no energy above 10 Hz (such as MSFR and ESPC). In addition, very stable spectral peaks are evident in some stations (such as IVCR, ESAL and HAGA) probably due to continuously active seismic noise sources. More interestingly, a general reduction in the amplitude of seismic noise at all the stations, even at the ones located in remote areas such as Mt. Etna flanks and the

Aeolian Islands, is observed following the enforcement of lockdown measures (on 11 March 2020; Figures 2-5). However, the amount of reduction, as well as the pattern of the investigated seismic RMS amplitude time series varies significantly in function of the station considered. For instance, the stations located close to towns and infrastructures such as busy roads, highways, industrial plants and agricultural areas (EFIU, ESML, HSRS, HPAC; Table 1), show the typical temporal pattern of the anthropogenic seismic noise with minima during the weekends and the night-time, and maxima during the weekdays

and the day-time (Figure A4; Lecocq et al., 2020; Xiao et al., 2020). Focusing on the frequency band 10-40 Hz, the amplitude noise reduction due to the lockdown measures reaches ~50% at EFIU station (Figure 5), which also shows a slight amplitude decrease (~10%) during the Christmas - New Year holidays. This station is close to towns, as well as to a busy highway called A18 (Figure A1 and Table 1). Other stations show less regular patterns with clear peaks interspersed throughout the time series, whose origin depends also in this case on the station considered. In the stations located in the Aeolian Islands (ISTR, ILLI,

IVCR, IFIL), the peaks are closely related to the ships rather than to road traffic or industrial activities (Figure A1 and A5, Table 1). The amplitude and rate of occurrence of those peaks also clearly decreased right after the implementation of the lockdown measures. Indeed, after March 2020 marine traffic was affected by a dramatic decrease at the global scale, that was particularly marked in the Mediterranean Sea (March et al., 2020; Figure 11b-d). Overall, the reduction in seismic noise in the band 10-40 Hz in the Aeolian Islands ranges between ~40% - 50% (Figure 5). At stations EMFS, ESPC, located on the flanks

of Mt. Etna, the anthropogenic seismic noise is mostly related to tourists' excursions, as both stations are located close to country roads used to bring tourists to the top of the volcano (Figure A1 and Table 1). These excursions were suspended on 9 March following the COVID-19 outbreak, leading to a decrease in the seismic noise in the band 10-40 Hz of 30% and 60% for ESPC and EMFS, respectively (Figure 5). Some notable increases in seismic amplitudes, visible in more than one stations at the same time, are not caused by human activities but rather due to bad weather conditions. For instance, the amplitude

increase on 25-26 March, visible at almost all the stations (Figures 2 and 3), was associated with bad weather conditions, which affected the whole Southern Italy. The amplitude of the anthropogenic seismic noise interestingly increased again at the end of April (Figures 3 and 4), a few days before the introduction of the first Presidential Decree, slightly releasing the lockdown measures (4 May 2020), and many days before the following Presidential Decree lifting those measures for many economic activities (18 May 2020).

As for the frequency content analysis, the spectral ratios show that the band characterised by anthropogenic seismic noise is strongly station-dependent (Figures 6 and 7). Indeed, the frequency with the maximum ratio value, coinciding with the frequency most affected by the anthropogenic noise, ranges from a few Hz (i.e. MSFR) to 20 Hz or even more (i.e. EFIU, ESML). Besides, the maximum ratio value, which indicates the amount of anthropogenic noise affecting the station, shows a fairly wide variability, from 1.5 (i.e. ESPC, HPAC) to 6 (EMFS). It is also noteworthy that some stations show a ratio lower

than 1 at low frequencies (<5 Hz), indicating that the seismic amplitude was higher during the lockdown than before. The stations on or around Mt. Etna (ESPC, ESAL, EFIU, AIO, EMFS, ESML), as well as the station on Stromboli island (ISTR), exhibit this behaviour due to the increase of volcanic tremor amplitude in both volcanoes during the second analysed time window (11 March – 1 April) with respect to the first one (1-20 February). Indeed, Mt. Etna and Stromboli volcanoes are characterised by continuous volcanic tremor, with energy mainly radiated in the band ~0.5-5.5 Hz (e.g. Cannata et al., 2010;

Falsaperla et al., 1998). In addition, the variability of the spectral ratio values below 1 Hz is not related to temporal changes in anthropogenic noise but rather to the variations in the amplitude of microseism, the most continuous and ubiquitous seismic signal on Earth, generated by ocean wave energy coupling with the Earth's ground (e.g. Longuet-Higgins 1950; Hasselmann, 1963; Ardhuin et al., 2015). Finally, the spectral ratio derived from stacking the spectral ratio plots of all the considered stations clearly shows that the frequency band most affected by the anthropogenic seismic noise (that is, the one showing the highest

ratio values) is above 10 Hz (Figure 8).

Concerning the comparison between the time series of seismic noise and human mobility, these are positively correlated, suggesting that seismic noise amplitude increases with increasing human mobility, with the exception of the residential visit changes provided by Google (Figure A2). Indeed, this category quantifies the change in duration of time spent at places of residence, which, unlike the other categories, increased during the lockdown period. We obtained a wide range of Spearman

correlation coefficients, whose absolute values range from 0.25 to more than 0.85 according to the considered station and mobility parameter (Figure 10). In particular, stations EFIU, ESML and HSRS, displaying the typical temporal pattern of the anthropogenic seismic noise with minima during the weekends and maxima during the weekdays, showed the highest correlation coefficients. The similarity between seismic noise and human mobility patterns is remarkably high at station EFIU characterized by correlation coefficients higher than 0.8 (Figures 9a and A2). For other stations (e.g. EMFS, HAGA, MUCR),

the correlation is not as clear. However, even if the correlation between seismic noise and human mobility is strongly station-dependent, p-values lower than 0.05 were obtained in all the comparisons (Figure A3), suggesting how the obtained Spearman correlation coefficients are significantly different from zero. This confirms that seismic data from all the considered stations contain plenty of information about human mobility. The seismic noise frequencies better correlated with human mobility data turned out to be above 10 Hz (maximum values are reached in the band 11 - 18 Hz; Figure 9b). This result, in line with the

information obtained by computing the ratios between the seismic spectra before and during the lockdown (Figure 8), shows how the seismic frequency band most affected by the human activities is above 10 Hz. As for the comparison between the time series of seismic noise at ILLI station in the band 10-40 Hz and ship traffic data of Lipari port, only daily overall gross tonnage

and the daily number of port calls of tanker showed p-values lower than 0.05 (with Spearman correlation coefficients equal to 0.18 and 0.25, respectively). In this case, the seismic noise frequencies better correlated with ship traffic data were below ~25 Hz. This analysis, focused on ship traffic data, highlight how particular types of boats (likely the ones with higher gross tonnage, among which the tankers) are mainly responsible of the seismic noise generation recorded close to ports. In addition, the frequency band most affected by such a seismic noise seems to be different (in particular lower) from the band most influenced by the other human activities (see Figures 9b and 11e).

Since the general reduction in the amplitude of seismic noise at all the stations in mid-March took place at the end of the winter - beginning of the spring and then when the meteorological conditions improved, such a decrease could also be interpreted as due to weather changes. Indeed, wind generates a broadband seismic noise with frequencies from ~0.5 Hz up to ~60 Hz (e.g. Bormann and Wielandt, 2013). However, we exclude such a possibility on the basis of: i) the increase of seismic noise amplitude, observed again at the end of April; ii) the correlation analysis results, that confirm how the amplitude reduction in mid-March is related to the decrease in human mobility.

## 4. Conclusions

The amplitude reduction of the anthropogenic seismic noise, due to the lockdown measures restricting the mobility of citizens during the COVID-19 pandemic, gave the opportunity to investigate in detail the characteristics of such an anthropogenic signal in Eastern Sicily.

We emphasize how the seismic amplitude decrease is visible even at stations located in remote areas, such as Mt. Etna volcano and the Aeolian Islands. The amount of the amplitude reduction, its temporal pattern and spectral content proved to be strongly station-dependent. As for the former, we found decreases of 30-60% in most of the considered stations, located close to towns and busy highways, as well as on the flanks of Mt. Etna where country roads are used to bring tourists on the top of the volcano, or in the Aeolian Islands following to ship traffic reduction. Regarding the temporal patterns of seismic noise amplitude, the stations installed close to towns or infrastructures (like busy roads and highways) showed the typical pattern of the anthropogenic seismic noise with minima during the weekends and the night-time, and maxima during the weekdays and the day-time. Other stations show less regular patterns with clear peaks interspersed throughout the time series. Concerning the spectral content, the frequency band most affected by anthropogenic seismic noise ranges from a few Hz to more than 20 Hz, depending on the station. On average, the frequencies above 10 Hz are the most influenced by anthropogenic seismic noise. We found that human mobility influenced the seismic noise mostly in frequencies above 10 Hz with remarkably high correlations between them observed at some stations. Furthermore, the comparison between seismic noise data acquired at ILLI station (located in Lipari; Aeolian Islands) and ship traffic data from Lipari port highlighted a significant correlation,

especially in case of ships with high gross tonnage. These results further confirm how seismic data, routinely acquired

worldwide mainly for seismic and volcanic surveillance, can also be used to monitor human mobility, especially during emergency periods such as the COVID-19 pandemic. As highlighted by Lindsey et al. (2020), seismic data also present the advantages over mobile phone-derived information of being natively anonymous and not affected by biases due to data sampling according to socio-economic class, age and region.

**Data availability**


The data that support the findings of this study are available on request from the corresponding author.

**Author contributions**

A.C., F.C. and T. L. designed the study. All the Authors analysed the seismic data. A.C., F.C., R.D.P., G.D.G. wrote the paper.

All the Authors interpreted the results and revised the article.

**Competing interests**

The authors declare that they have no conflict of interest.

**Acknowledgements**


We are indebted to the technicians of the INGV-OE for enabling the acquisition of seismic data. We also sincerely acknowledge all the essential workers who, placing themselves at risk, are taking care of the sicks, assisting the communities, and serving in any way during this COVID-19 pandemic. This work was funded by CHANCE project, II Edition, Università degli Studi di Catania. Figure 1 was drawn by readhgt.m Matlab® function (Copyright 2020, François Beauducel;

https://it.mathworks.com/matlabcentral/fileexchange/36379-readhgt-import-download-nasa-srtm-data-files-hgt) and shows topography data provided by NASA Shuttle Radar Topography Mission (SRTM). We would like to thank Stephen Hicks, Kasper van Wijk and an anonymous reviewer for their great help to improve this manuscript. Ship traffic data from Lipari port was provided by FleetMon.com, JAKOTA Cruise Systems GmbH, Rostock, Germany (data provided on 10 October 2020).

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

# Appendix A: additional figures

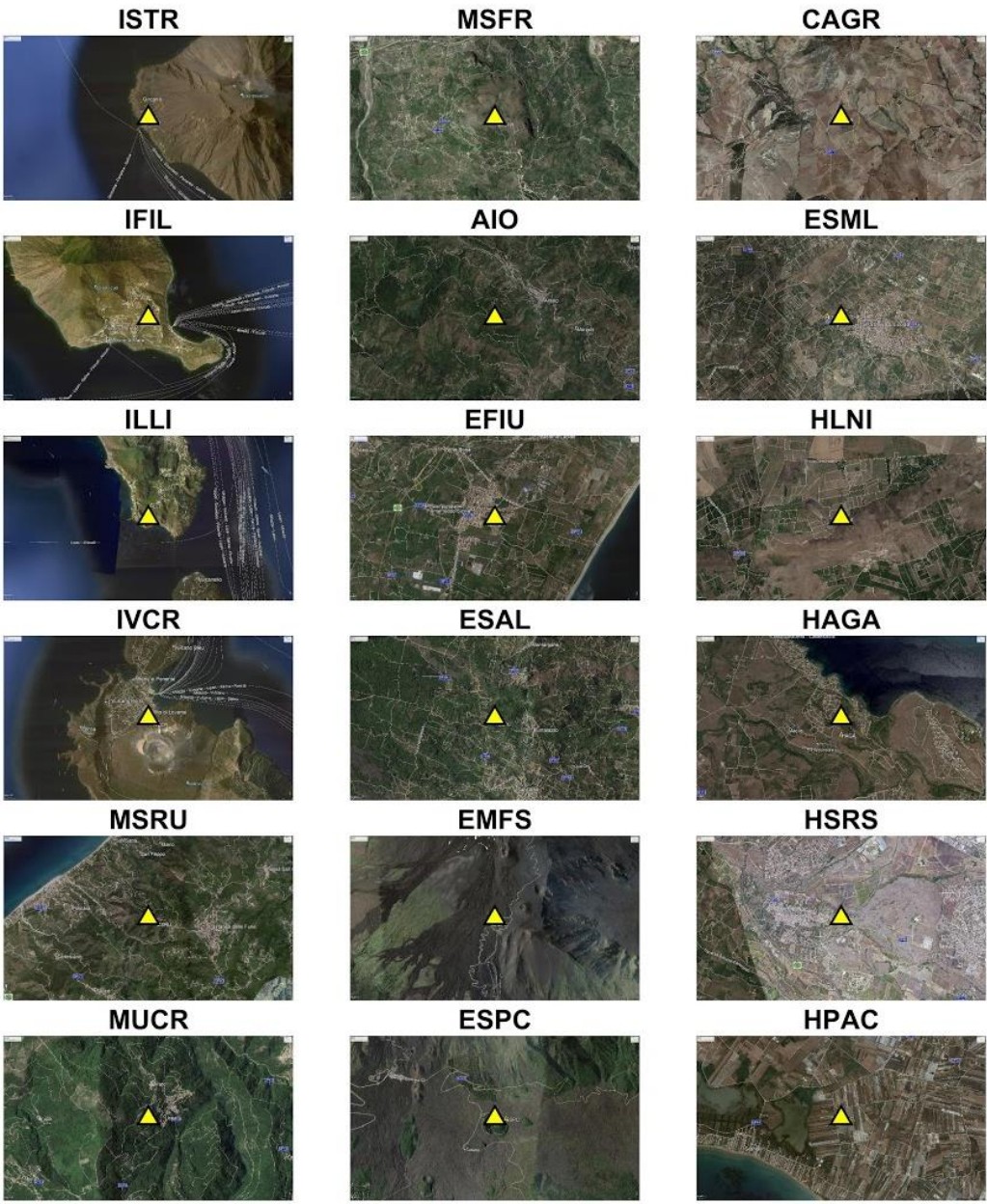

**Figure A1. Photos from © Google Earth of the installation site areas of the seismic stations (yellow triangles) used in this work. The width of the area shown in each picture is ~5 km. The stations are sorted by decreasing latitude from upper left to bottom right.**

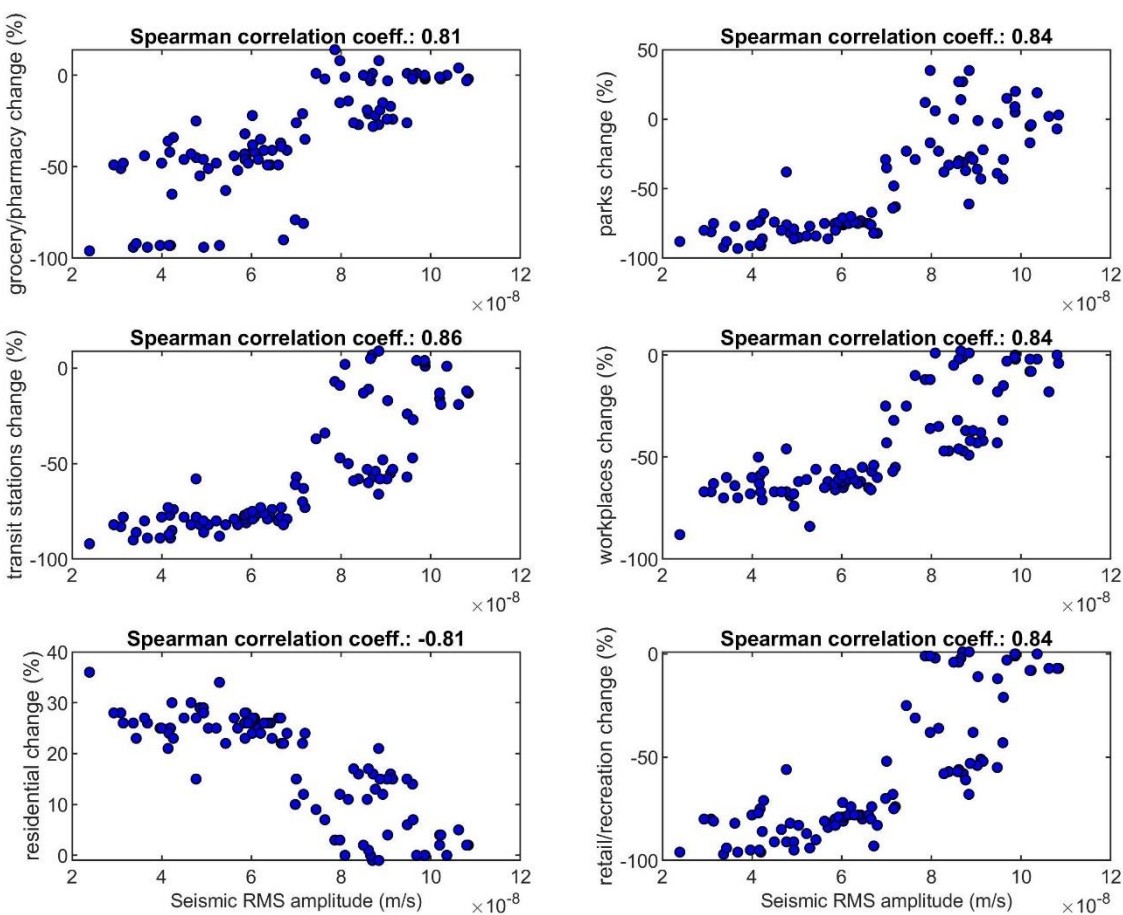

**Figure A2. Cross-plots showing the seismic RMS amplitude of EFIU station in the band 10-40 Hz in the x-axis and the different categories of human mobility from Google in the y-axis from 15 February to 20 May 2020. The values of the Spearman correlation coefficient, computed between the time series plotted in the graph, are shown in the title.**


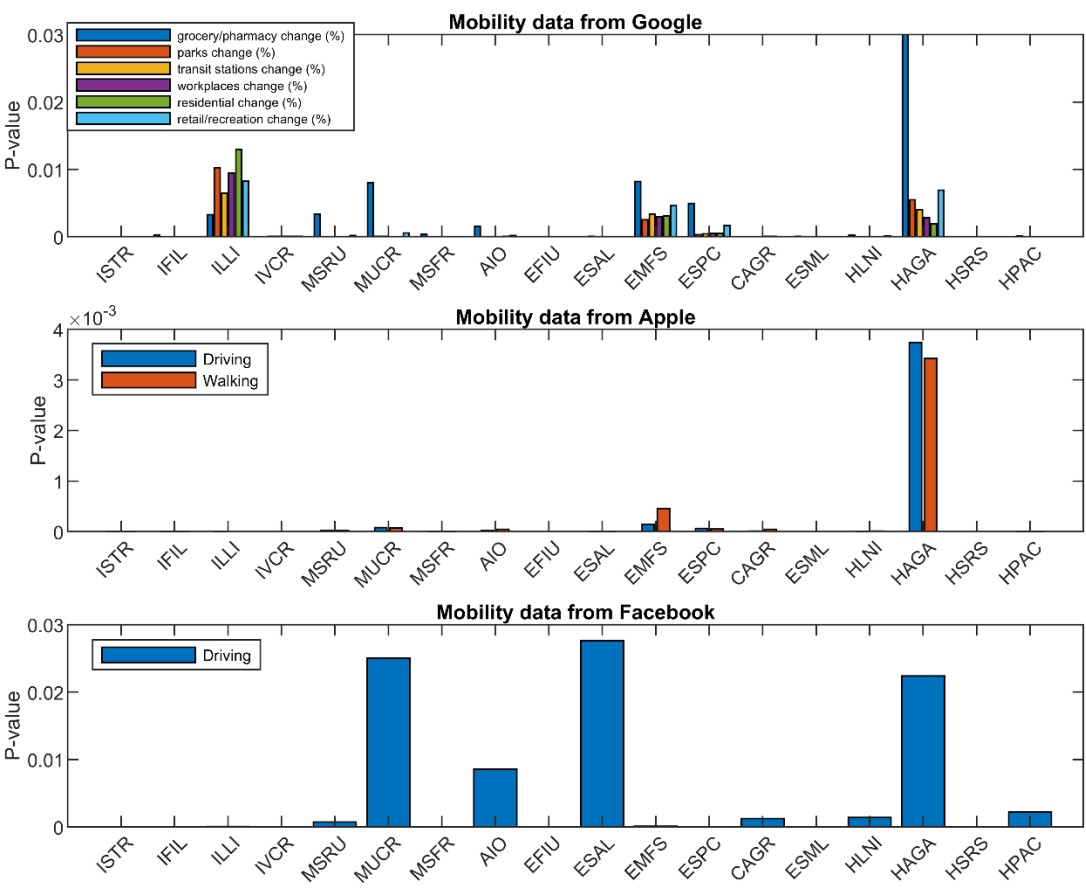

**Figure A3. Probability values (p-values), obtained by the Spearman correlation analysis performed between seismic RMS amplitude at the different stations and the human mobility parameters, as provided by Google, Apple and Facebook.**

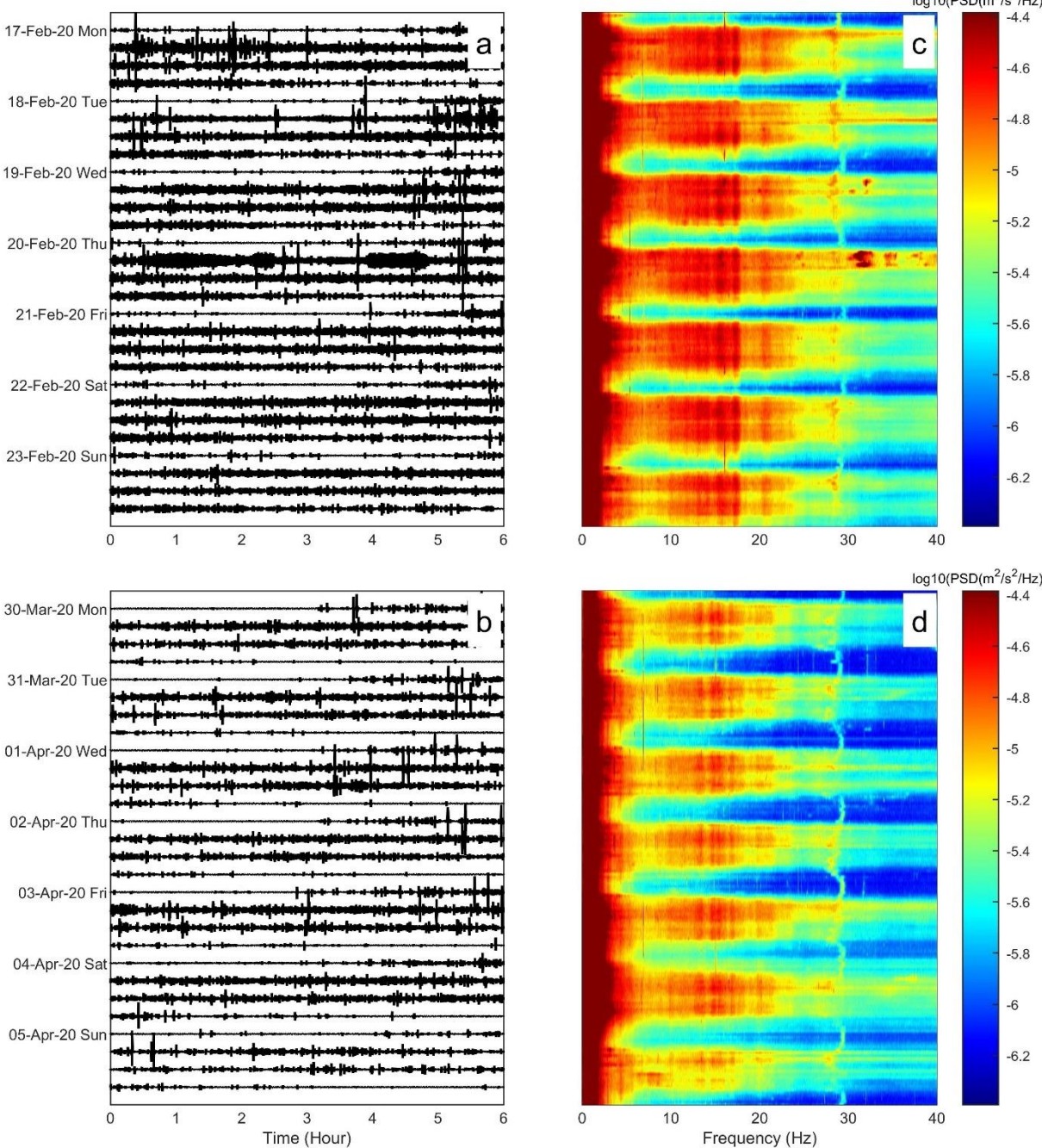


**Figure A4.** (a) Seismogram showing the seismic signal recorded by the vertical component of the station EFIU and filtered in the band 10-40 Hz during the week 17-23 February 2020 and (c) the corresponding spectrogram of the non-filtered signal. (b) Seismogram showing the seismic signal recorded by the vertical component of the station EFIU and filtered in the band 10-40 Hz during the week 30 March - 5 April 2020 and (d) the corresponding spectrogram of the non-filtered signal.


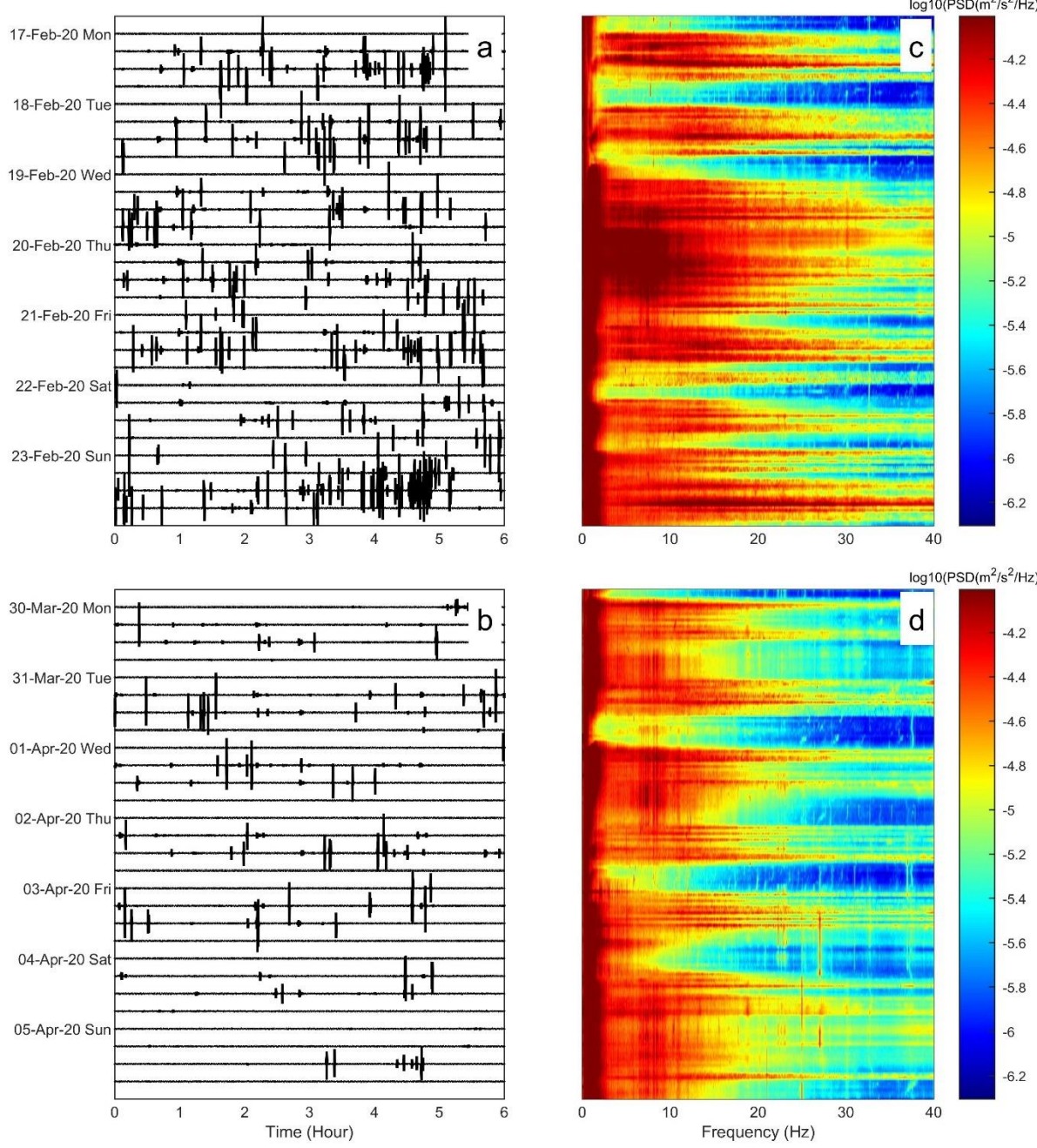

**Figure A5. (a) Seismogram showing the seismic signal recorded by the vertical component of the station ILLI and filtered in the band 10-40 Hz during the week 17-23 February 2020 and (c) the corresponding spectrogram of the non-filtered signal. (b) Seismogram showing the seismic signal recorded by the vertical component of the station ILLI and filtered in the band 10-40 Hz during the week 30 March - 5 April 2020 and (d) the corresponding spectrogram of the non-filtered signal.**