# Peer review of "Seismic evidence of the COVID-19 lockdown measures: a case of study from Eastern Sicily (Italy)"

_Solid Earth, 2020_

## Referee Comment (RC1) · Stephen Hicks (Referee) · 1 Oct 2020

**1. SUMMARY**

This manuscript by Cannata et al. analyses the effect of COVID-19 lockdowns on seismic noise in Sicily, Italy. Although this effect has been reported globally, this study is unique because uses a fairly dense regional seismic network to view the higher-order features of the anthropogenic lockdown signal and its spectral characteristics. The study finds quite a heterogeneous lockdown response, even on a relatively small island. Most interestingly perhaps, it is also found that the anthropogenic noise reduction might also have allowed for more detection of seismic arrivals from seismic events.

I enjoyed reading this manuscript. The writing is very clear and contains minimal errors.

[Figure]

I congratulate the authors on a very nice study.

Overall, the manuscript is in excellent shape, and aside from some minor comments detailed below, it should be close to publication-quality. My most significant comment is that I think the description of the seismicity detection results should be expanded a little.

2. MINOR GENERAL COMMENTS

1) The analysis of the earthquake detections is interestingly, yet somewhat disappointingly short. I recommend a few things:

- First of all, this aspect of the paper is not yet mentioned in the abstract, so I would add some mention of it there.

- Second, I would recommend perhaps not having a separate methods subsection on the earthquake detection. I would move this short description of the detection system along with the results to a separate subsection of the Discussion called something like "Implications of the lockdown for detection of seismicity". - Finally, I think is much scope for further analyses of the seismicity detection changes. For example, for Figure 11, I could expect that if the improve the detectability of seismic phases during lockdown is robust, then it should be seen most clearly for seismic events occurring during the daytime. How does the correlation look if you only plot seismic events from during the daytime (e.g 0600-1800)? Also, if you were to assess a Gutenberg-Richter relationship and to compare pre- and during-lockdown, would you be able to infer a lower completeness magnitude? Finally, are you able to determine which stations had more P-picks during the lockdown, if so, were these the noisier stations, e.g. "EFIU"?

2) Is there any specific reason why your frequency analysis only goes up to 30 Hz, when your stations were sampling data to 100 Hz, so possibly allowing you to get close to 50 Hz? It might be interesting to see what his happening with the anthropogenic seismic wavefield at higher frequencies.

3. MINOR SPECIFIC COMMENTS

- L35: "Between 8-11 March, the entire country was put under lockdown (Gatto et al., 2020)". This phrasing makes it sound like the country was on lockdown for 3 days between the 8 and 11 March. Please rephrase, including the approximate total length of lockdown.

- L55: I guess it would be good to use the opportunity in this paragraph to state why your study is different and complementary to the existing COVID-19 seismic noise studies. I guess yours is the first study that uses a fairly dense network from a local area in which lockdown restrictions were imposed uniformly. So, it gives us the opportunity to view higher-resolution details of the anthropogenic noise field (e.g. how the anthropogenic noise field propagates, site effects, frequency effects, etc.), with a uniform lockdown and independent of potential cultural variations.

L65: You mention the seismometer instrument type, but it would be good to describe the station installation styles and environments given that you are looking at a local-scale case study. Are all stations deployed in subsurface vaults? Or is there a more variable installation style? Are some stations located in populated areas, or are they in as remote regions as possible? Or is the installation style quite mixed over the network?

L170: I find this sentence a bit confusing: "The correction was performed by dividing the number of phases by the fraction of seismic data acquired by the network during the day when the earthquake took place, with respect to the data which would have been recorded in case of full operating state of the network (Figure 11b).". Does that mean the y-axis of Figure 11b is essentially a percentage value? Could you maybe please clarify this?

- Figure 1: Some of the station text labels are quite small, overlapping, and so are hard to read. Please increase the font size and edit the label positions to make sure they do not overlap. -> Please also include a small inset map for readers who may not know

where exactly Sicily is :) -> It might also be useful to include some topographic shading to emphasise the position and flanks of Mt. Etna.

- Figure 2: -> If the paper is printed on A4 paper, some of the text labels could be very small. Maybe consider increasing each subplot size and reduce the whitespace between subplots? -> The "LD" label is very hard to see. Maybe increase the font and put this in a semi-transparent box. -> The x-axis tick intervals are a bit random. Maybe just show the 1st day of each month for clarity.

- Figure 3: -> The "LD" label is very hard to see. Maybe consider increase the font and put this in a semi-transparent box.

- Figure 4: -> What is the order of the stations on the y-axis? If these are in no particular order then maybe using alphabetic order might be useful so that readers can easily crosscheck the station results with other figures. -> The "LD" label is very hard to see. Maybe increase the font and put this in a semi-transparent box.

- Figure 5: -> Some of the station text labels are quite small, overlapping, and so are hard to read.

- Figure 6: -> If the paper is printed on A4 paper, some of the text labels could be very small.

Figure 7: -> If the paper is printed on A4 paper, some of the text labels could be very small.

Figure 9: -> The figure resolution is very low so I cannot read the text labels in the legend.

Figure 11: -> Change the y-axis labels from "# picking" to "Number of P-picks".

[Figure]

---

## Referee Comment (RC2) · Anonymous Referee #2 · 2 Oct 2020

This contribution describes the effect of quieting following the COVID19 lockdown measures on the noise level in a regional seismic network located around the Etna, Sicily, Italy. The subject is of interest, in particular in the framework of the "Social seismology" SE Special issue.

The paper is well-written, the structure is in general well shaped and the figures are of good quality (although many labels should be enlarged). Therefore, I think that the manuscript deserves to be published in SE after minor to moderate revision.

However, there are a number of point that, in my opinion, should be reworked in the final version of the manuscript.

The most valuable contribution of the manuscript is documenting that a seismic noise

reduction can be observed in areas far from large cities, where human activity still affect the background seismic noise via ship transit, touristic excursions etc.. Even some stations which seems to be installed at remote places reflect the decrease of activity following lockdown. I think that this point has to be highlighted through the manuscript and in particular in the abstract and conclusions.

In general, the manuscript makes a good job in presenting the data, but tends to be too concise in the interpretation part. Sections 2.2 and 2.3 are merely descriptive of the results presented in the corresponding figures. The reader has to wait till section 3 to learn something on the information included in the figures. I propose to include here the discussion on the differences observed between sites, the tentative origin of noise at each site etc. included now at Section 3.

Figure 6 and 7 provide essentially the same information that Figs 2 and 3, presented in a different way. I will appreciate a comment on which are the advantages of each kind of representation. Are there features only observed these representations and not in the RMS or spectra?? If yes, it will be interesting to comment. Otherwise, the figures can be seen as redundant

Regarding the comparison with mobility data, I think that the message that seismic data is consistent with other data is best passed using a graphic as that presented at Suppl. Fig A2 than using just correlation coefficients. I suggest to start the section using a new figure composed by the a) subplot of Suppl fig. A2 and the submitted Fig. 10. The submitted Fig 9 will move to Fig 10. In this way the reader will first see an example of correlation between RMS and mobility data for specific station, then see the overall correlation and finally see the differences between stations and mobility data.

The discussion on Spearman correlation and t-test and p-values is unclear. I think that the original Figure 9 has to be used shown the station with good or poor correlation with mobility data, in some graphic, easy to interpret way.

In the discussion (line 220) it is stated that only the ESAL/Facebook correlation does

not match the criteria, but, in my opinion, some of the stations (EFIU, HSRS, ESML, HPAC) clearly show a good correlation between seismic RMS and quietening, while for the rest, the correlation is less clear. This point should be clearly stated, noting that the relationship seismic noise/mobility is not always clear. In the Conclusions section it is correctly stated that the effect is strongly station-dependent. I think that this dependency should be better described here. In any case I would enhance the fact that, even for stations with poor correlation there are evidences of changes in the seismic noise values.

I don't understand Suppl Fig A3; P-values are in the order of 0.005 for Google, 0.0002 for Apple and 0.02 for Facebook. Are those order of magnitude differences realistic?? The authors state the p-value of 0.05 is considered sufficient to reject the null hypothesis; while this particular number is chosen?

Noise level variations related to ship activity or touristic excursions is interesting and not often described. I suggest to give more weight to this funny observation.

The section on the improvement on detection capability has a large potential interest, but it is not really developed here. In the main text, the authors just describe Figure 11 and the final discussion includes just a sentence on this subject. If the authors decide to keep the section, a significant improvement will be needed. Figure 11 shows that the number of pickings increase during lockdown, but the relevant information will be if more small magnitude events are detected or if the hypocentral determination is improved during lockdown. This analysis should be taken carefully, taking into account the epicentral distance of the events detected in each period, the occurrence of swarms/aftershocks that could perturb the comparison etc. Otherwise, a better option will be to keep the detection improvement discussion for a next paper focused on this subject.

Other points.

L. 31: Not sure that references to papers dealing with pharmacology are needed here

L. 65: The actual location setting of each location is hard to see in the small size screenshots in Supp Fig A1. I propose to summarize in this paragraph the different setting of the stations; how many are in towns, near roads, in small islands, in open nature etc Also a comment on the kind of installation used in each case will be useful; different installation types (vault, buried, building basement, insulation system etc) could affect the sensibility to human activity noise.

Line 85: The authors should explain why they decided to use the 10-30 Hz band. I suggest to use the submitted Figure 8 to justify this choice.

Fig 2 and 3: I will appreciate more conventional time labels (p.e. 1st and 15th of each month)

Fig 2: The bars marking lockdown beginning is difficult to see

Fig. 6 and 7: Labels are too small

Fig. 10: The correlation is calculated between mobility data and a mean RMS profile using all the available data? Please clarify

L. 130: Using the Spearman correlation coefficient is really justified ? Are the results using Pearson really different ??

L. 140. The reference to the critical p-value level to reject the null hypothesis should be better explained. As stated above, the numbers in the y-axis of Supp Fig A3 seems very different

---

## Referee Comment (RC3) · Kasper van Wijk (Referee) · 7 Oct 2020

"Seismic evidences of the COVID-19 lockdown measures: Eastern Sicily case of study" analyzes the data from the seismic network on Sicily during the lockdown. I am attaching an annotated pdf with smaller comments, mainly with some writing-related suggestions. In terms of the science, the analysis is thorough, and could be published in its current form. However, I wanted to propose something for the authors/editors to consider. To me, Figure 11 is the most exciting result: an increase in detection levels for earthquakes during the lockdown. I would provide more info (and data to show the increased S/N!) on this, and have a more focused build-up to this result, and have maybe less of the first 10 figures, as most of those observations were already reported in other settings in the existing published literature on this topic. If the authors agree, the ab-

stract and conclusions should also highlight this result with quantitative information on this enhanced detection level.

Finally, I was wondering if weather data is available for the region? I say this, because it may be that winds could shake trees and buildings affecting seismic noise, even in the 10+ Hz band. If you agree, a correlation between wind speed (for example) and seismic noise levels may help build the case that enhanced detection level of earthquakes is due to anthropogenic quieting during the COVID-19 lockdown on Sicily.

Sincerely,

Kasper van Wijk

Please also note the supplement to this comment:
https://se.copernicus.org/preprints/se-2020-136/se-2020-136-RC3-supplement.pdf

**Supplement:**

[revised manuscript text omitted]

---

## Author Comment (AC1) · 27 Oct 2020

**Reply to Reviewer #1 (Stephen Hicks)**

1. SUMMARY

This manuscript by Cannata et al. analyses the effect of COVID-19 lockdowns on seismic noise in Sicily, Italy. Although this effect has been reported globally, this study is unique because uses a fairly dense regional seismic network to view the higher-order features of the anthropogenic lockdown signal and its spectral characteristics. The study finds quite a heterogeneous lockdown response, even on a relatively small island. Most interestingly perhaps, it is also found that the anthropogenic noise reduction might also have allowed for more detection of seismic arrivals from seismic events. I enjoyed reading this manuscript. The writing is very clear and contains minimal errors.

I congratulate the authors on a very nice study. Overall, the manuscript is in excellent shape, and aside from some minor comments detailed below, it should be close to publication-quality. My most significant comment is that I think the description of the seismicity detection results should be expanded a little.

> *We thank the reviewer for the very positive comments.*

2. MINOR GENERAL COMMENTS

1) The analysis of the earthquake detections is interestingly, yet somewhat disappointingly short. I recommend a few things:

- First of all, this aspect of the paper is not yet mentioned in the abstract, so I would add some mention of it there.

- Second, I would recommend perhaps not having a separate methods subsection on the earthquake detection. I would move this short description of the detection system along with the results to a separate subsection of the Discussion called something like "Implications of the lockdown for detection of seismicity".

- Finally, I think is much scope for further analyses of the seismicity detection changes. For example, for Figure 11, I could expect that if the improve the detectability of seismic phases during lockdown is robust, then it should be seen most clearly for seismic events occurring during the daytime. How does the correlation look if you only plot seismic events from during the daytime (e.g 0600-1800)? Also, if you were to assess a Gutenberg-Richter relationship and to compare pre- and during-lockdown, would you be able to infer a lower completeness magnitude? Finally, are you able to determine which stations had more P-picks during the lockdown, if so, were these the noisier stations, e.g. "EFIU"?

> *We tried to repeat the analysis by considering only earthquakes recorded during day-time and weekdays, but the number of events became very low (25 and 85, during and before lockdown, respectively), to be statistically significant. In addition, we also evaluated the Gutenberg-Richter relationship separately for earthquakes, taking place during and before lockdown, and we did not note any significant changes in the completeness magnitude, equal to 1.6 in both cases:*

[Figure]

*Hence, these further analyses did not allow to get more robust results. Probably, this topic would need a more in-depth analysis. Following the advice of the reviewer #2, we decided to delete this section regarding the detection improvement and keep it for a next narrower, more focused study.*

2) Is there any specific reason why your frequency analysis only goes up to 30 Hz, when your stations were sampling data to 100 Hz, so possibly allowing you to get close to 50 Hz? It might be interesting to see what his happening with the anthropogenic seismic wavefield at higher frequencies.

*> Thanks for your advice. We extended our analysis up to 40 Hz, and consequently most of the figures/analyses have been updated. We could not go beyond 40 Hz, because of the digitizer anti-aliasing low-pass filter with cut-off frequency around 40-45 Hz (it depended on the station taken into account).*

3. MINOR SPECIFIC COMMENTS

- L35: "Between 8-11 March, the entire country was put under lockdown (Gatto et al., 2020)". This phrasing makes it sound like the country was on lockdown for 3 days between the 8 and 11 March. Please rephrase, including the approximate total length of lockdown.

*> Done, thanks.*

- L55: I guess it would be good to use the opportunity in this paragraph to state why your study is different and complementary to the existing COVID-19 seismic noise studies. I guess yours is the first study that uses a fairly dense network from a local area in which lockdown restrictions were imposed uniformly. So, it gives us the opportunity to view higher-resolution details of the anthropogenic noise field (e.g. how the anthropogenic noise field propagates, site effects, frequency effects, etc.), with a uniform lockdown and independent of potential cultural variations.

*> We added a sentence in that paragraph, highlighting the peculiarity of this study. Thanks for your advice.*

- L65: You mention the seismometer instrument type, but it would be good to describe the station installation styles and environments given that you are looking at a local scale case study. Are all stations deployed in subsurface vaults? Or is there a more variable installation style? Are some stations located in populated areas, or are they in as remote regions as possible? Or is the installation style quite mixed over the network?

> *The installation style is pretty uniform, all the considered stations are installed in shallow vaults (depth ~1.5 m) made of concrete. As for the site conditions (in terms of possible anthropogenic seismic sources), it is variable: some are close to towns or highways, others near agricultural areas, others in small islands or on the flanks of Mt. Etna. All these information have been added in the manuscript, as well as in the new Table 1.*

- L170: I find this sentence a bit confusing: "The correction was performed by dividing the number of phases by the fraction of seismic data acquired by the network during the day when the earthquake took place, with respect to the data which would have been recorded in case of full operating state of the network (Figure 11b).". Does that mean the y-axis of Figure 11b is essentially a percentage value? Could you maybe please clarify this?

> *As mentioned above, this topic would need a more in-depth analysis. Following the advice of the reviewer #2, we decided to delete this section regarding the detection improvement and keep it for a next narrower, more focused study.*

- Figure 1: Some of the station text labels are quite small, overlapping, and so are hard to read. Please increase the font size and edit the label positions to make sure they do not overlap. -> Please also include a small inset map for readers who may not know where exactly Sicily is :) -> It might also be useful to include some topographic shading to emphasise the position and flanks of Mt. Etna.

> *Done.*

- Figure 2: -> If the paper is printed on A4 paper, some of the text labels could be very small. Maybe consider increasing each subplot size and reduce the whitespace between subplots? -> The "LD" label is very hard to see. Maybe increase the font and put this in a semi-transparent box. -> The x-axis tick intervals are a bit random. Maybe just show the 1st day of each month for clarity.

> *Done.*

- Figure 3: -> The "LD" label is very hard to see. Maybe consider increase the font and put this in a semi-transparent box.

> *Done.*

- Figure 4: -> What is the order of the stations on the y-axis? If these are in no particular order then maybe using alphabetic order might be useful so that readers can easily crosscheck the station results with other figures. -> The "LD" label is very hard to see. Maybe increase the font and put this in a semi-transparent box.

> *The stations are sorted by decreasing latitude (it is indicated in the caption). We increased the font size of "LD" to make it more visible.*

- Figure 5: -> Some of the station text labels are quite small, overlapping, and so are hard to read.

> *Done.*

- Figure 6: -> If the paper is printed on A4 paper, some of the text labels could be very small.

> *Done.*

Figure 7: -> If the paper is printed on A4 paper, some of the text labels could be very small.

> *Done.*

Figure 9: -> The figure resolution is very low so I cannot read the text labels in the legend.

> *We increased the resolution, and changed a bit the figure based on the advices of the reviewer #2.*

Figure 11: -> Change the y-axis labels from "# picking" to "Number of P-picks".

> *As above mentioned, we deleted this section regarding the increased detectability of earthquakes, and then also the related figure.*

---

## Author Comment (AC2) · 27 Oct 2020

**Reply to Reviewer #2**

This contribution describes the effect of quieting following the COVID19 lockdown measures on the noise level in a regional seismic network located around the Etna, Sicily, Italy. The subject is of interest, in particular in the framework of the "Social seismology" SE Special issue. The paper is well-written, the structure is in general well shaped and the figures are of good quality (although many labels should be enlarged). Therefore, I think that the manuscript deserves to be published in SE after minor to moderate revision. However, there are a number of point that, in my opinion, should be reworked in the final version of the manuscript.

*> We thank the reviewer for the positive comments.*

The most valuable contribution of the manuscript is documenting that a seismic noise reduction can be observed in areas far from large cities, where human activity still affect the background seismic noise via ship transit, touristic excursions etc.. Even some stations which seems to be installed at remote places reflect the decrease of activity following lockdown. I think that this point has to be highlighted through the manuscript and in particular in the abstract and conclusions.

*> Thanks for your advice, we added a couple of sentences about that in abstract, Results and discussion and conclusions sections.*

In general, the manuscript makes a good job in presenting the data, but tends to be too concise in the interpretation part. Sections 2.2 and 2.3 are merely descriptive of the results presented in the corresponding figures. The reader has to wait till section 3 to learn something on the information included in the figures. I propose to include here the discussion on the differences observed between sites, the tentative origin of noise at each site etc. included now at Section 3.

*> I see your point; however, we cannot write results in the "Materials and methods" section. Indeed, by putting only data and analysis descriptions in the "Materials and methods" section, the readers, not interested in the technical details, can skip such a part of the manuscript and still get all the information about the main paper results.*

Figure 6 and 7 provide essentially the same information that Figs 2 and 3, presented in a different way. I will appreciate a comment on which are the advantages of each kind of representation. Are there features only observed these representations and not in the RMS or spectra?? If yes, it will be interesting to comment. Otherwise, the figures can be seen as redundant.

*> Actually, figures 6 and 7 are necessary to explore the spectral content of the anthropogenic seismic noise and contain information not present (or at least not that evident) in both spectrograms and RMS amplitude time series. Hence, in our opinion, such figures cannot be considered redundant. Among the figures you cited,*

*probably the one we commented less was Figure 2, showing the spectrograms. Hence, we added some comments about these at the beginning of the section 3.*

Regarding the comparison with mobility data, I think that the message that seismic data is consistent with other data is best passed using a graphic as that presented at Suppl. Fig A2 than using just correlation coefficients. I suggest to start the section using a new figure composed by the a) subplot of Suppl fig. A2 and the submitted Fig. 10. The submitted Fig 9 will move to Fig 10. In this way the reader will first see an example of correlation between RMS and mobility data for specific station, then see the overall correlation and finally see the differences between stations and mobility data.

> *Done, thanks for your advice.*

The discussion on Spearman correlation and t-test and p-values is unclear. I think that the original Figure 9 has to be used shown the station with good or poor correlation with mobility data, in some graphic, easy to interpretate way.

> *We changed a bit the discussion of these data. We hope that it is clearer now.*

In the discussion (line 220) it is stated that only the ESAL/Facebook correlation does not match the criteria, but, in my opinion, some of the stations (EFIU, HSRS, ESML, HPAC) clearly show a good correlation between seismic RMS and quietening, while for the rest, the correlation is less clear. This point should be clearly stated, noting that the relationship seismic noise/mobility is not always clear. In the Conclusions section it is correctly stated that the effect is strongly station-dependent. I think that this dependency should be better described here. In any case I would enhance the fact that, even for stations with poor correlation there are evidences of changes in the seismic noise values.

> *As above mentioned, we have modified the discussion on these data. In particular, we have clearly written that correlation between seismic noise and human mobility is strongly station-dependent, some stations show very good correlations, others less. In any case, the p-value is lower than 0.05 in all the comparisons, suggesting how the obtained Spearman correlation coefficients are significantly different from zero. In addition, we performed again the analysis by using seismic RMS amplitude in the band 10-40 Hz (in place of 10-30 Hz). By doing so, the correlation ESAL/Facebook now also matches the criterion p<0.05.*

I don't understand Suppl Fig A3; P-values are in the order of 0.005 for Google, 0.0002 for Apple and 0.02 for Facebook. Are those order of magnitude differences realistic?? The authors state the p-value of 0.05 is considered sufficient to reject the null hypothesis; while this particular number is chosen?

> *The p-value threshold of 0.05 means that the probability, that the result of the statistical test is due to chance alone, is less than 5%, so it would occur once out of 20 times the study is repeated. The value of 0.05*

*is a commonly accepted significance level used for this statistic test. We added a couple of sentences about that in the section 2.3. As for the obtained differences in p-values, yes, they are realistic.*

Noise level variations related to ship activity or touristic excursions is interesting and not often described. I suggest to give more weight to this funny observation.

*> Agreed. Indeed, ship traffic data of Lipari port has been obtained and compared with seismic data acquired by ILLI station (installed in Lipari Island at about 2.7 km from the port). Hence, a new figure and several sentences have been added regarding this topic in sections 2.3 and 3. Furthermore, a couple of sentences about the fact that the seismic noise amplitude reduced even in stations installed in remote places have been added in abstract, Results and discussion and conclusions sections.*

The section on the improvement on detection capability has a large potential interest, but it is not really developed here. In the main text, the authors just describe Figure 11 and the final discussion includes just a sentence on this subject. If the authors decide to keep the section, a significant improvement will be needed. Figure 11 shows that the number of pickings increase during lockdown, but the relevant information will be if more small magnitude events are detected or if the hypocentral determination is improved during lockdown. This analysis should be taken carefully, taking into account the epicentral distance of the events detected in each period, the occurrence of swarms/aftershocks that could perturb the comparison etc. Otherwise, a better option will be to keep the detection improvement discussion for a next paper focused on this subject.

*> We totally agree with you. This topic would need a more in-depth analysis. Following your advice, we decided to delete this section regarding the earthquake detection improvement and keep it for a next narrower, more focused study.*

Other points.

L. 31: Not sure that references to papers dealing with pharmacology are needed here

*> The aim of lockdown measures, which influenced so unexpectedly the seismic signals, was just to slow down the COVID-19 epidemic to give more time to the pharmacological research to find a cure and/or a vaccine against COVID-19. This is the reasoning behind the references to papers dealing with pharmacology. So, in our opinion, they are needed here.*

L. 65: The actual location setting of each location is hard to see in the small size screenshots in Supp Fig A1. I propose to summarize in this paragraph the different setting of the stations; how many are in towns, near roads, in small islands, in open nature etc Also a comment on the kind of installation used in each case will

be useful; different installation types (vault, buried, building basement, insulation system etc) could affect the sensibility to human activity noise.

> *We added a description of the installation, as well as the Table 1 summarizing the site conditions in terms of possible anthropogenic seismic sources.*

Line 85: The authors should explain why they decided to use the 10-30 Hz band. I suggest to use the submitted Figure 8 to justify this choice.

> *Exactly. We selected that frequency band according to both spectral ratio and spectral correlation with the human activity. However, reviewer #1 suggested to extend the analyses at higher frequencies, hence we chose the band 10-40 Hz, and performed again most of the analyses. We added a sentence in section 2.2 to justify our choice.*

Fig 2 and 3: I will appreciate more conventional time labels (p.e. 1st and 15th of each month)

> *Done.*

Fig 2: The bars marking lockdown beginning is difficult to see.

> *Done.*

Fig. 6 and 7: Labels are too small.

> *Done.*

Fig. 10: The correlation is calculated between mobility data and a mean RMS profile using all the available data? Please clarify.

> *We have rephrased the part of the manuscript, describing this analysis (Section 2.3). It should be clearer now.*

L. 130: Using the Spearman correlation coefficient is really justified? Are the results using Pearson really different??

> *As we explained in the Section 2.3, since we do not know whether the relationship between seismic noise and mobility data is linear or not, the Spearman correlation analysis is more recommended than the Pearson correlation.*

L. 140. The reference to the critical p-value level to reject the null hypothesis should be better explained. As stated above, the numbers in the y-axis of Supp Fig A3 seems very different.

*> Yes, they are. We checked the computations, and they are correct. In addition, we added a couple of sentences about the critical p-value level in the section 2.3.*

---

## Author Comment (AC3) · 27 Oct 2020

**Reviewer #3 (Kasper van Wijk)**

"Seismic evidences of the COVID-19 lockdown measures: Eastern Sicily case of study" analyzes the data from the seismic network on Sicily during the lockdown. I am attaching an annotated pdf with smaller comments, mainly with some writing-related suggestions. In terms of the science, the analysis is thorough, and could be published in its current form. However, I wanted to propose something for the authors/editors to consider.
> *We really thank the reviewer for the positive comments, as well as for the very helpful suggestions reported in the attached pdf.*

To me, Figure 11 is the most exciting result: an increase in detection levels for earthquakes during the lockdown. I would provide more info (and data to show the increased S/N!) on this, and have a more focused build-up to this result, and have maybe less of the first 10 figures, as most of those observations were already reported in other settings in the existing published literature on this topic. If the authors agree, the abstract and conclusions should also highlight this result with quantitative information on this enhanced detection level.
> *Following the advices of the other two reviewers, we repeated the earthquake detection analysis by considering only earthquakes recorded during day-time and weekdays, but the number of extracted events was very low (25 and 85, during and before lockdown, respectively), to be statistically significant. In addition, we also evaluated the Gutenberg-Richter relationship separately for earthquakes, taking place during and before lockdown, and we did not note any significant changes in the completeness magnitude, equal to 1.6 in both cases. Hence, we think that this topic would need a more in-depth analysis. Following the advice of the reviewer #2, we decided to delete this section regarding the detection improvement and keep it for a next narrower, more focused study.*

Finally, I was wondering if weather data is available for the region? I say this, because it may be that winds could shake trees and buildings affecting seismic noise, even in the 10+ Hz band. If you agree, a correlation between wind speed (for example) and seismic noise levels may help build the case that enhanced detection level of earthquakes is due to anthropogenic quieting during the COVID-19 lockdown on Sicily.
> *Actually, the period preceding the lockdown falls in winter, while the lockdown mainly in spring, so worse weather conditions (as well as the corresponding more intense seismic noise of meteorological origin) are expected in the former. However, in the following plot, we show daily wind speed data as recorded by a meteorological station installed in the Catania airport (red line is the moving average over 10 days). Such data do not show so evident changes in March 2020, to make you think that the observed seismic noise decrease could be due to meteorological variations.*

[Figure]

*In addition, the seismic noise amplitude started increasing again at the end of April, suggesting that the previous decrease cannot be due to variable weather conditions, but rather to anthropic activities. Finally, the correlation analysis between seismic data and human mobility confirms that the amplitude reduction is related to the decrease in anthropic activities. We added some sentences about this at the end of section 3.*

***Other comments in the attached pdf:***

where does this very precise, but not round number come from? why not 82 seconds?

*> Such a number is due to the fact that spectral analysis by FFT requires power of two for the number of data points; the signals are acquired at 100 Hz, and then 81.92 s corresponds with 8192 points ($2^{13}$).*

how about a line for when the LD ended?

*> We added a line in Figures 2, 3, 4 and 9a on 4 May 2020, when the first Presidential Decree, slightly releasing the lockdown measures, was issued.*

you may have discussed this later in the paper, but how did weather affect this result? The island stations in the North seem to have a large reduction. Could this be that one of the periods had more wave action than the other?

*> Sea wave action should create seismic noise (microseism) at low frequencies, below 1 Hz. In Figure 5 we showed the percent change of seismic RMS amplitude in the band 10-40 Hz. In addition, the seismograms shown in Figure A5(a) clearly show that the decrease in noise amplitude is not due to variations in a continuous signal (as microseism should be) but rather to the reduction in the occurrence rate of amplitude transients that in the Aeolian Islands are associated with ship activities. Finally, we added a reference, confirming that during the first period of COVID-19 pandemic marine traffic was affected by a dramatic decrease at the global scale, as well as ship traffic data of Lipari port, that correlate fairly well with seismic noise variations in ILLI station.*